# Object Tracking Based on Satellite Videos: A Literature Review

**Zhaoxiang Zhang** [1] , **Chenghang Wang** [1] , **Jianing Song** [2,*] and **Yuelei Xu** [1]

1    Unmanned System Research Institute, Northwestern Polytechnical University, Xi'an 710129, China; zhangzhaoxiang@nwpu.edu.cn (Z.Z.); 201806060228@sust.edu.cn (C.W.); xuyuelei@nwpu.edu.cn (Y.X.)
2    Department of Electrical and Electronic Engineering, University of London, London ECV1 0HB, UK
\*    Correspondence: jianing.song@city.ac.uk

**Abstract:** Video satellites have recently become an attractive method of Earth observation, providing consecutive images of the Earth's surface for continuous monitoring of specific events. The development of on-board optical and communication systems has enabled the various applications of satellite image sequences. However, satellite video-based target tracking is a challenging research topic in remote sensing due to its relatively low spatial and temporal resolution. Thus, this survey systematically investigates current satellite video-based tracking approaches and benchmark datasets, focusing on five typical tracking applications: traffic target tracking, ship tracking, typhoon tracking, fire tracking, and ice motion tracking. The essential aspects of each tracking target are summarized, such as the tracking architecture, the fundamental characteristics, primary motivations, and contributions. Furthermore, popular visual tracking benchmarks and their respective properties are discussed. Finally, a revised multi-level dataset based on WPAFB videos is generated and quantitatively evaluated for future development in the satellite video-based tracking area. In addition, 54.3% of the tracklets with lower Difficulty Score (DS) are selected and renamed as the Easy group, while 27.2% and 18.5% of the tracklets are grouped into the Medium-DS group and the Hard-DS group, respectively.

**Keywords:** satellite video; traffic target tracking; ship tracking; typhoon tracking; fire tracking; ice motion tracking; deep learning

## 1. Introduction

Object tracking is a hot topic in computer vision and remote sensing, and it typically employs a bounding box that locks onto the region of interest (ROI) when only an initial state of the target (in a video frame) is available [1,2]. Thanks to the development of satellite imaging technology, various satellites with advanced onboard cameras have been launched to obtain very high resolution (VHR) satellite videos for military and civilian applications. Compared to traditional target tracking methods, satellite video target tracking is more efficient in motion analysis and object surveillance, and has shown great potential applications in spying on enemies [3], monitoring and protecting sea ice [4], fighting wildfires [5], and monitoring city trafficking [6], which traditional target tracking cannot even approach.

Recent research has shown an increasing interest in traditional video-based target tracking, with numerous algorithms proposed for accurate tracking in computer vision. Methods that utilize generative models [7–10] or discriminant models [11–17] can be divided into two categories. The generative model-based target tracking can be thought of as a search problem, in which the object area in the current frame is modeled and the most similar region is chosen as the predicted location in the next frame. In contrast, discriminant models regard object tracking as a binary classification problem and have attracted much attention due to their efficiency and robustness [18]. A classifier is used and trained for discriminant models, with the attributes of the object and background labeled as positive and negative samples in the current frame. In the following frame, the classifier is used to identify the foreground, and the results are updated.

There are three major modules in general visual-based object tracking [19–21], which are: (1) target representation scheme, defining a target that is of interest for further analysis, such as vehicles or ships; (2) search mechanism, estimating the state of the target objects; (3) model update step, updating the target representation or model to account for appearance variations. Because of the different features of remote sensing images, satellite video tracking has confronted several issues compared with traditional object tracking tasks or unmanned aerial vehicle (UAV)-based aerial image tracking. The challenges of employing object-tracking technology in satellite video datasets are listed as follows [22]:

- Small foreground size compared with the background: The width and height of high-resolution satellite video are usually more than 2000 pixels, while the interested target only takes up about 0.01% of the whole video frame pixels or even less. The large-size background expands the searching region of classic tracking algorithms while decreasing tracking performance. Furthermore, tracking targets of tiny size have fewer features and are similar to the environment, resulting in less tracking robustness and a large tracking error.
- Low video frame rate: Because of onboard hardware limitations, the frame rate of satellite video is typically low, resulting in significant movement of the object targets between frames and further influencing tracking prediction and model update. For example, if the target is abruptly stopped, obscured, or shifted, existing tracking systems can easily miss it.
- Sudden illumination change: Because the satellite video collection is collected at a high altitude in space, the light and atmospheric refraction rate vary with the orbital satellite's motion, which could result in an abrupt change in frame lighting. The difference in light has a significant impact on the performance and accuracy of object tracking.

Traditional visual tracking methods utilize various frameworks, such as discriminative correlation filters (DCF) [23], Siamese network (SN) [24], tracking-by-detection (TBD) [25,26], and silhouette tracking. However, due to the constraints mentioned above, these approaches cannot deliver good performance in satellite video tracking. As a result, new research has updated and altered old methods to deal with satellite video tracking.

Previous works have reviewed the object detection methods based on general videos and aerial videos. Refs. [1,27,28] investigated traditional methods in terms of classical object and motion representation by examining the pros and cons either systematically or experimentally, or both. Refs. [29,30] divided handcrafted and deep visual trackers into correlation filter (CF) trackers and non-CF trackers and then employed a classification based on architectures and tracking mechanisms. Ref. [31] systematically investigated deep-learning-based visual tracking methods, benchmark datasets, and evaluation metrics. Ref. [31] analyzed the deep learning (DL)-based methods from six aspects: network architecture, network exploitation, network training for visual tracking, network objective, network output, and the exploitation of CF advantages. Ref. [32] reviewed object tracking methods aiming at aerial surveillance videos, starting from the development history and current research institutions, and then focusing on the UAVs-based tracking methods by providing detailed descriptions of the common frameworks that contain ego-motion compensation, representative tracking algorithms, and object TBD.

Table 1 gives a brief characteristic of previous reviews or surveys. Compared with our work, we put special focus on both traditional and DL-based techniques for target tracking using satellite remote sensing data with the targets varying from artificial objects (traffic objects and ships) and natural objects (typhoon, fire, and ice motion).The main contributions of this paper are summarized as follows:

(1) Various satellite video-based visual tracking technologies are classified based on their monitoring goals, tracking network training (online or offline tracking), and network tracking. The motivations and contributions of various tracking systems for satellite video targets are discussed. This is, to the best of our knowledge, the first document that reviews the key concerns and solutions to satellite video-based tracking problems.

(2) By analyzing their fundamental properties, the existing satellite video benchmark datasets are compared and analyzed.

(3) Based on the Wright Patterson Air Force Base (WPAFB) dataset, a revised multi-level dataset with manual annotation is constructed, and quantitative and qualitative experimental evaluations for the aforementioned dataset are presented.

The rest of the paper is organized as follows: Section 2 introduces the methodology and overview of the proposed review process. Sections 3–6 review the tracking framework and algorithm in terms of five different tracking target (traffic object, ship, typhoon, fire, and ice), respectively. Section 8 analyzes the common benchmark datasets, with further discussion and a novel multi-level dataset based on WPAFB dataset. Finally, Section 9 concludes the paper.

**Table 1.** Characteristic of previous reviews and surveys.

| Ref. | Year | Target | Data | Technique |
|------|------|--------|------|-----------|
| [1] | 2006 | general object tracking | general videos | traditional techniques |
| [33] | 2020 | video object tracking | general videos | traditional & DL |
| [16] | 2021 | social object tracking | social media | CF-based method |
| [34] | 2013 | traffic monitoring | UAV data | traditional technique |
| [31] | 2019 | visual tracking | UAV data | DL technique |
| [35] | 2021 | traffic detection and tracking | UAV data | DL technique |
| [36] | 2022 | pedestrians/cars tracking | UAV data | Siamese networks |
| [5] | 2020 | wildfire observation | UAV data | traditional technique |
| [37] | 2021 | fire detection and analysis | satellite multi-spectral data | traditional technique |
| [38] | 2014 | Ship Surveillance | space-borne SAR and AIS | traditional technique |

## 2. Methodology and Overview of Taxonomy in Satellite Video Tracking Methods

In this study, related works from the last ten years are identified by the Web of Science (WoS) database and Google scholar search engine with the keywords such as *satellite video tracking*, *aerial video tracking*, *remote sensing image and tracking*, *satellite video*, and *remote sensing images*. Reviewed works are restricted to peer-reviewed documents, including journals and conference papers, to ensure the authenticity and quality of the outcomes.

A comprehensive review of the satellite video-based visual tracking methods is presented in terms of three aspects: tracking targets, tracking training methods, and tracking architecture. From a high-level perspective, the tracking targets are divided into artificial targets and natural targets, where vehicles, ships, trains, and planes are examples of artificial ones, and typhoons, fire, and ice are the category of the natural target. Due to their wide range of social significance and economic value, these seven objects have drawn much attention from researchers, and massive works have been proposed in recent years.

Nevertheless, some other satellite applications are not discussed in the following sections but are only listed here because of little published data on the specific applications. These applications include but are not limited to wild animal tracking [39], cloud tracking, tree defoliation tracking [40], low-salinity pool tracking [41], deep convective cloud tracking [42], crop phenology tracking [43], etc. Furthermore, traffic object tracking is one of most interest within the field of satellite video-based visual tracking due to its promising application potential and performance. We, therefore, divide the traffic object tracking algorithms into two training approaches: online tracking and offline tracking. The mainstream online tracking methods include optical flow-based methods, TBD-based methods, CF-based methods, and DL-based methods according to their architectures. The ship tracking algorithms are divided into image-based tracking approaches and multimodality-based tracking approaches based on the different model inputs. As for the typhoon target, the tracking models are categorized as convolutional neural network (CNN)-based methods and recurrent neural network (RNN)-based methods according to their model structure. Meanwhile, some of the fire and ice target tracking methods are based on traditional meth-

ods, while other tracking approaches are based on DL. The proposed taxonomy of satellite video-based visual tracking methods is illustrated in Figure 1.

In the following sections, not only are state-of-the-art satellite video-based visual tracking systems classified, but also the motives and contributions of those approaches are discussed, as well as helpful thoughts on future developments.

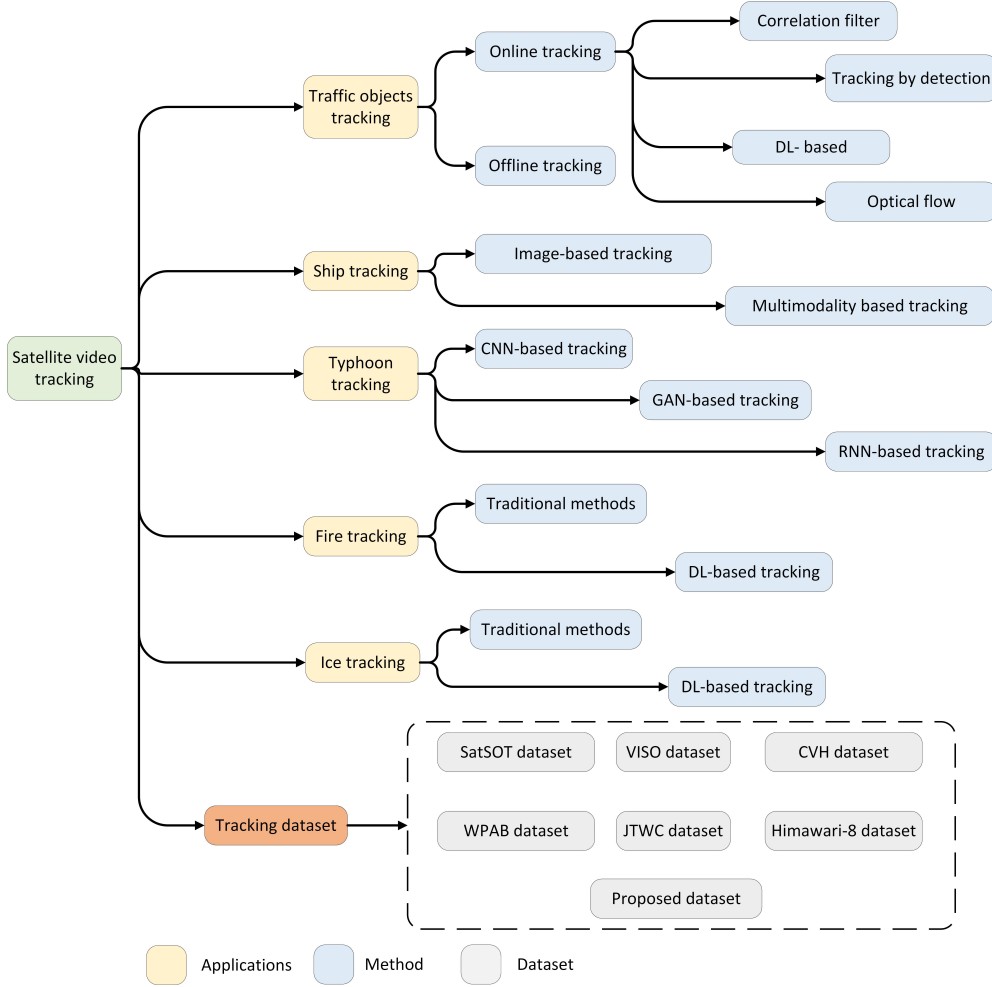

**Figure 1.** The tree diagram of satellite video-based visual tracking algorithms.

## 3. Traffic Object Tracking

In this section, the traffic object tracking methods are reviewed under two headings, which are online tracking methods and offline tracking methods. Furthermore, the online tracking methods are grouped into four broad types: CF-based, TBD, DL-based, and optical flow-based methods. Finally, discussions on reviewed traffic tracking methods are delivered.

### 3.1. Online Tracking Methods

As presented in Section 1, tracking using satellite video has confronted many challenges compared with traditional object tracking tasks because of the characteristics of satellite video data, such as large scene size, small target size, few features, and similar background. Thus, various tracking architectures have been proposed to deal with the above challenges. The mainstream solutions (Figure 2) to satellite video-based tracking consist of the optical flow-based method, the CF-based method, the DL-based method, and the TBD-based method.

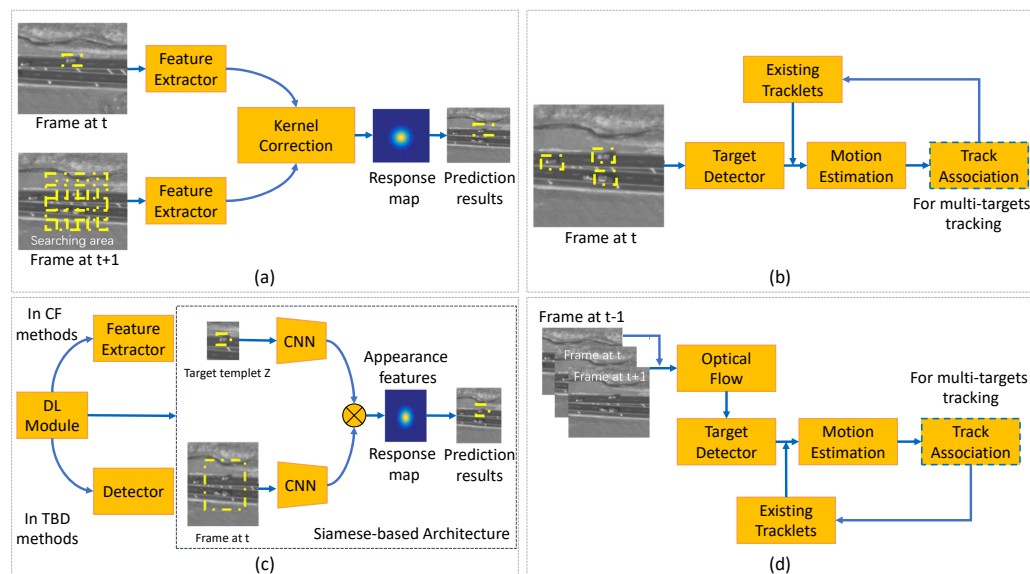

**Figure 2.** General architecture sketch of online tracking methods for traffic objects. (**a**) CF-based tracking methods; (**b**) TBD methods; (**c**) DL-based methods; (**d**) optical flow-based methods.

### 3.1.1. Correlation Filter-Based Tracking Methods

The CF has yielded promising results in optical tracking tasks and is one of the most popular tracking algorithms in satellite videos. However, the CF-based tracker achieves poor results because the size of each target compared with the entire image is too small. Several improved strategies are proposed herein for taking advantage of the CF to gain a better tracking performance. Table 2 summarizes specific CF-based tracking methods. As shown in Table 2, recent CF-based tracking methods for traffic objects are of three kinds: (1) kernel correlation tracker (KCF) with multi-frame case; (2) KCF for target motion case; and (3) KCF aided by kernel adaptation. The general pipeline of the CF-based tracking methods is depicted in Figure 2a.

**Table 2.** Summary of various CF-based traffic objects tracking methods.

| Methods | Ref. | Year | Description |
|---|---|---|---|
| KCF + multi-frames | [6] | 2017 | KCF with the three-frame-difference |
| | [44] | 2022 | KCF with Multi-feature fusion |
| KCF + target motion | [45] | 2019 | Improved discriminative CF for small objects tracking |
| | [46] | 2019 | High-speed CF-based tracker for object tracking |
| | [47] | 2019 | KCF embedded with motion estimations |
| | [48] | 2021 | Feature fusion, position compensation, local object region |
| KCF + kernel adaptation | [49] | 2018 | VCF using velocity feature and inertia mechanism |
| | [50] | 2019 | Hybrid KCF with histogram of oriented gradient |
| | [51] | 2021 | Rotation-adaptive CF |
| | [52] | 2022 | Rotation-adaptive CF with motion constraint |
| | [53] | 2022 | Spatial-Temporal regularized CF with interacting multiple model |
| | [54] | 2022 | Kernelized CF with color-name features |

In 2017, Ref. [6] presented a new object tracking method by taking advantage of the KCF and the three-frame-difference method to deal with satellite videos. The integrated model combined the shape information provided by the KCF tracker and the change information from the three-frame-difference method into the final tracking results. Three videos that described the conditions of Canada, Dubai, and New Delhi were introduced, with the target of moving trains and cars. The image sizes of these three videos were $3840 \times 2160$ pixels for both the first and second videos and $3600 \times 2700$ pixels for the

third one. The average center location error (CLE) and the average overlap score were 11 pixels and 71%, respectively. Later, Ref. [44] presented a KCF embedded method that fused multi-feature fusion and compensates motion trajectory to track fast-moving objects in satellite videos. The contributions of the suggested algorithm were multifold. First, a multi-feature fusion strategy was proposed to describe an object comprehensively, which was challenging for the single-feature approach. Second, a subpixel positioning method was developed to calculate accurate object localization that was further used to improve the tracking accuracy. Third, the adaptive Kalman filter (AKF) was introduced to compensate for the KCF tracker results and reduce the object's bounding box drift, solving the moving object occlusion problem. Compared to the KCF algorithm, the algorithm improved the tracking accuracy and the success rate with over 17% and 18% on average.

In 2019, Ref. [45] developed an improved discriminative CF for small object tracking in satellite videos. Instead of employing a change detection tracking model, the authors first proposed a spatial mask to promote the CF to give different contributions depending on the spatial distance. The Kalman filter (KF) was then applied to predict the target position in the large and analogous background region. Next, the integrated strategy was applied to combine the improved CF tracker and pose estimation algorithm. The proposed model was implemented on the Chang Guang Satellite dataset with an image resolution of $3840 \times 2160$ pixels. The authors calculated success rate, precision, and frame per second (FPS) measurement indicators to evaluate the performance, achieving the result of 0.725, 0.96, and 1500, respectively. Comparing with other video tracking methods, including Channel and Spatial Reliability Tracker (CSRT) [55], Efficient Convolution Operator Tracker (ECOT) [56], long-term correlation tracker [57], and KCF models, the proposed method performed best.

Later, a high-speed CF-based tracker was derived by [46] for object tracking in satellite videos. The authors introduced the global motion characteristics of the moving vehicle target to constrain the tracking process. By integrating the position and velocity KF, the trajectory of the moving target was corrected. The tracking confidence module (TCM) was proposed to couple the KF and CF algorithms tightly, in which the confidence map of the tracking results was obtained by the CF and passed to the KF for a better prediction. The authors cropped the satellite videos of SkySat-1 and Jilin-1 into nine short sequences, which contained 31 moving objects in total, and then applied their method to the cropped satellite videos. Five metrics, namely, expected average overlap (EAO), accuracy, robustness, average overlap, and FPS, were used to evaluate the capability of the proposed method for object tracking, with the results of 0.7205, 0.71, 0.00, 0.7053, and 1094.67, respectively. Thus, the introduced technique was verified to be effective and fast for real-time vehicle tracking in satellite videos. Similarly, Ref. [47] studied a KCF embedded with motion estimations to track satellite video targets. The authors developed an innovative motion estimation (ME) algorithm combining the KF and motion trajectory to average and mitigate the boundary effects of KCF. An integrated strategy based on motion estimation was proposed to solve the problem of tracking failure when a moving object was partially or completely occluded. The experimental dataset consisted of 11 videos with a resolution of 1 m from the Jilin-1 satellite constellation. The area under curve (AUC), CLE, overlap score, and FPS measurement indicators were utilized to evaluate the tracking performance, which is 72.9, 94.3, 96.4, and 123, respectively. Compared with other object tracking methods, the developed model gained the best results. Furthermore, Ref. [48] proposed an improved KCF to track the object in satellite videos. The improvements of the proposed algorithm were: (1) fusing the different features of the object, (2) proposing a motion position compensation algorithm by combining the KF and motion trajectory, and (3) extracting the local object region for normalized cross-correlation matching. Thus, the algorithm was able to track the moving object in satellite video with high accuracy effectively.

Differing from the above feature-kernel-based tracking methods, Ref. [49] considered the extremely inadequate quality of target features in satellite videos. The authors designed a velocity correlation filter (VCF) by employing the velocity feature and inertia mechanism

to construct a KCF for satellite video target tracking. The velocity feature, with the high discriminative ability and inertial mechanism, could help to detect moving targets and prevent model drift in satellite videos. The experiment results showed that the AUC scores in precision and success plots of the proposed method reached 0.941 and 0.802, respectively. Moreover, the presented tracker had a favorable speed compared to other state-of-the-art methods, running at over 100 FPS. Later, Ref. [50] designed a hybrid kernel correlation filter tracker for satellite video tracking. This approach integrated the optical flow features with the histogram of oriented gradient and obtained competitive results. Similarly, Ref. [51] presented a rotation-adaptive CF tracking algorithm to address the problem caused by the rotation objects. The authors proposed an object rotation estimation method to keep the feature map stable for the object rotation and achieved the capability of estimating the change in the bounding box size. Ref. [52] decoupled the rotation and translation motion patterns and developed a novel rotation adaptive tracker with motion constraints. Experiments based on the Jilin-1 satellite dataset and International Space Station dataset demonstrated the superiority of the proposed method. To handle the occlusion problem during the satellite tracking, Ref. [53] developed a spatial-temporal regularized correlation filter algorithm with interacting multiple models. The authors utilized the interacting multiple models to predict the target position when the target is occluded. Similarly, Ref. [54] designed a kernelized correlation filter based on the color-name feature and Kalman prediction. Experiment results on Jilin-1 datasets show that the proposed algorithm has stronger robustness for several complex situations such as rapid target motion and similar object interference.

### 3.1.2. TBD Methods

Detection association strategy in computer vision is one of the popular methods for multi-target tracking [58]. By assigning detected candidates of each frame into trackers, the motion interpolation is utilized to retrieve the short-term missing detected candidates. This type of tracker is anointed TBD. However, unique characteristics of satellite videos, including low frame rate, less discriminative appearance information, and lacking color features, bring further challenges to current TBD methods. Table 3 summarizes specific references of the TBD methods and Figure 2b shows the general pipeline of this kind of method, in which four types are further divided on the basis of the tracking features. These are motion feature-based, hyperspectral image-based, graph-based, and discriminative-based TBD methods.

**Table 3.** Summary of TBD methods for traffic objects tracking.

| Method | Ref. | Year | Description |
|---|---|---|---|
| Motion feature-based tracking | [59] | 2017 | Local context tracker |
| | [60] | 2021 | SFMFT for multiple moving objects |
| Hyperspectral image-based tracking | [61] | 2016 | Study hyperspectral and spatial domain information |
| | [62] | 2017 | Real-time HLT method |
| Graph-based tracking | [63] | 2010 | Unified relation graph approach from road structure |
| Discriminative-based tracking | [64] | 2017 | Bayesian classification with motion smoothness constraint |
| | [65] | 2019 | Multi-morphological cue based discrimination strategy |
| | [66] | 2020 | TBD with filter training mechanism |

In the tracking by detection method, the detection models play an important role in enhancing the tracking performance. Classic detectors such as YOLO [67], CenterNet [68], and CornerNet [69] have been applied for object tracking. For example, Ref. [70] unified Cornernet and data association to achieve a better speed-accuracy trade-off for multi-object tracking while eliminating the extra feature extraction process.

To reduce the dependency on motion detection of frame differencing and appearance information, Ref. [59] introduced a local context tracker. In their method, the local context

tracker explored spatial relations for the target to avoid unreasonable model deformation in the next frame. The merged detection results in the detection association were explicitly handled, and short tracks were produced by associating hypotheses. The track association fused the results from two trackers and updated the "track pool" to improve the tracking performance. The designed model was tested on WPAFB Sequence and Rochester Sequence containing 410 and 44 tracks. Multiple metrics, namely, Recall, Precision, and the number of breaks per track (B/T), were introduced to analyze the performance, with the results of 0.606, 0.99, and 0.159, showing that the proposed method outperformed the state-of-the-art methods in satellite video-based tracking.

Aiming at tracking multiple moving objects, Ref. [60] proposed the slow feature and motion feature-guided multi-object tracking (SFMFT) method by using the slow features and motion features. Specifically, the authors developed a nonmaximum suppression (NMS) module to assist the object detection by utilizing the sensitivity of slow feature analysis to the changed pixels. This method reduced the amount of static false alarms and supplemented missed objects, further improving the recall rate by increasing the confidence score of the correctly detected object bounding boxes. The superiority of the proposed method was evaluated and demonstrated with three satellite videos.

On the other hand, Ref. [61] presented a real-time tracking method that exploits the hyperspectral and spatial domain information, aiming to reduce false alarm tracking rates. In their method, the individual feature map was computed for each hyperspectral band and then fed to an adaptive fusion method. Therefore, the fusion map with reduced noise could help to detect the targets from the background pixels efficiently. The CLIFF-2007 dataset with 0.3 cm Ground Sampling Distance (GSD) and 50 tracking targets was used to evaluate the suggested techniques. In terms of the track purity and target purity, the proposed hyperspectral feature-based method outperformed the Red-Green-Blue (RGB) only features, with the results of 64.37 and 57.49, respectively. Compared with their previous work, Ref. [62] designed an improved real-time hyperspectral likelihood maps-aided tracking (HLT) method. An online generative target model is proposed and revised for the tracking system of a target detection segment, considering the hyperspectral channels ranging from visible to infrared wavelengths. An adaptive fusion method is proposed to combine likelihood maps from multiple bands of hyperspectral imagery into one single more distinctive representation. The experimental outcomes indicate that the proposed model is able to track the traffic targets accurately.

Instead of exploring the tracking features from the targets, Ref. [63] developed a unified relation graph approach to explore vehicle behavior models from road structure and regulate object-based vertex matching in multi-vehicle satellite videos. The proposed vehicle travel behavior models generated additional constraints for better matching scores. Moreover, the authors utilized three-frame moving object detection to initialize vehicle tracks and a tracking-based target indicator to reduce miss-detection and refine the target location. The dataset used for evaluation was collected by a single camera covering a $1 \, km^2$ area with a frame rate of $1 \, Hz$. The Multiple Object Tracking Accuracy (MOTA) [71] was introduced as a metric for the accuracy assessment and was 0.85 achieved by the proposed method, thereby indicating satisfactory results for satellite video tracking. The model could be further improved by preparing extra high-quality satellite videos with tracking labels.

The above-discussed methods can be seen as graph-based methods, which explore the target movement model according to their graph features. There is another well-studied strategy for object tracking based on the discriminative method. In 2017, Ref. [64] proposed a Bayesian classification considering the motion smoothness constraint to track vehicles in satellite videos. The authors introduced the gray level similarity feature to describe the likelihood of the target with the assumption of motion smoothness, and the posterior probability was used to identify the tracking target position. Additionally, a KF was introduced to enhance the robustness of tracking processing. The SkySat and Jilin-1 satellite dataset were applied to evaluate the proposed model, showing the superiority and potential of the model for object tracking from remote sensing imagery. Later, Ref. [65] presented

a modified detection-tracking framework to identify and track small moving vehicles in satellite sequences. An original detection algorithm was developed based on local noise modeling and exponential probability distribution. After detection, a discrimination strategy based on the multi-morphological cue was designed to further identify correct vehicle targets from noises. The suggested method was employed in the Chang Guang Satellite dataset. F1 score, recall, precision, Jaccard Similarity, MOTA, and Multiple Object Tracking Precision (MOTP) were calculated to assess classification performance, with the results of 0.71, 63.06, 81.04, 0.55, 0.46, and 0.52, respectively. Furthermore, Ref. [66] exploited the circulant structure of TBD with Kernels, and established a filter training mechanism for the target and background to improve the discrimination ability of the tracking algorithm. Tracking experiments with nine sets of Jilin-1 satellite videos showed competitive performance with targets under weak feature attributes.

### 3.1.3. DL-Based Tracking Methods

CNN models have achieved significant success in many vision tasks, which inspires researchers to explore their capabilities in tracking problems. State-of-the-art CNN-based trackers have made remarkable progress toward this goal [56,72–74], showing more robust than traditional methods with a large training dataset. However, DL-based trackers need to adapt to satellite videos due to the challenges of the satellite videos-based target tracking issues discussed in Section 1. Figure 2c illustrates the general pipeline of DL-based methods, where the DL modules are utilized in the Siamese architecture to extract the appearance features. Moreover, the DL modules can be introduced into CF-based methods and TBD methods, running as the feature extractor and feature detector. For instance, Ref. [75] utilized CNN to extract hyperspectral domain features and a kernel-based CF dealing with the satellite video tracking problem.

A SN is an CNN-based approach that applies the same weights while working in tandem on two different input vectors to compute comparable output vectors, which is typically utilized for comparing similar instances in different type sets. Thus, it is a natural idea to apply the SN in the object tracking task [74]. In 2019, Ref. [76] constructed a fully convolutional SN with shallow-layer features to retrieve fine-grained appearance features for space-borne satellite video tracking (Figure 3a). Predicting attention combined Gaussian Mixture Model (GMM), and KF was utilized to deal with tracking target occlusion and the obscure problem. The proposed method was validated by three high-resolution satellite videos quantitatively, which outperformed the state-of-the-art tracking methods with an FPS of 54.83. Similarly, a deep Siamese network (DSN) incorporating an interframe difference centroid inertia motion (ID-CIM) model was proposed in Ref. [77], in which the ID-CIM mechanism was proposed to alleviate model drift. The DSN inherently included a template branch and a search branch and extracted the features from these two branches. A Siamese region proposal network was then employed to obtain the target position in the search branch. Meanwhile, [78] investigated a lightweight parallel network with a high spatial resolution to locate the small objects in satellite videos, namely, the Hign-resolution Siamese network (HRSiam). A pixel-level refining model based on online moving object detection and adaptive fusion was proposed to enhance the tracking robustness in satellite videos. By modeling the video sequence in time, the HRSiam detected the moving targets in pixels with the advantage of tracking and detecting. The authors reported that their proposed HRSiam achieved state-of-the-art tracking performance while running at over 30 FPS.

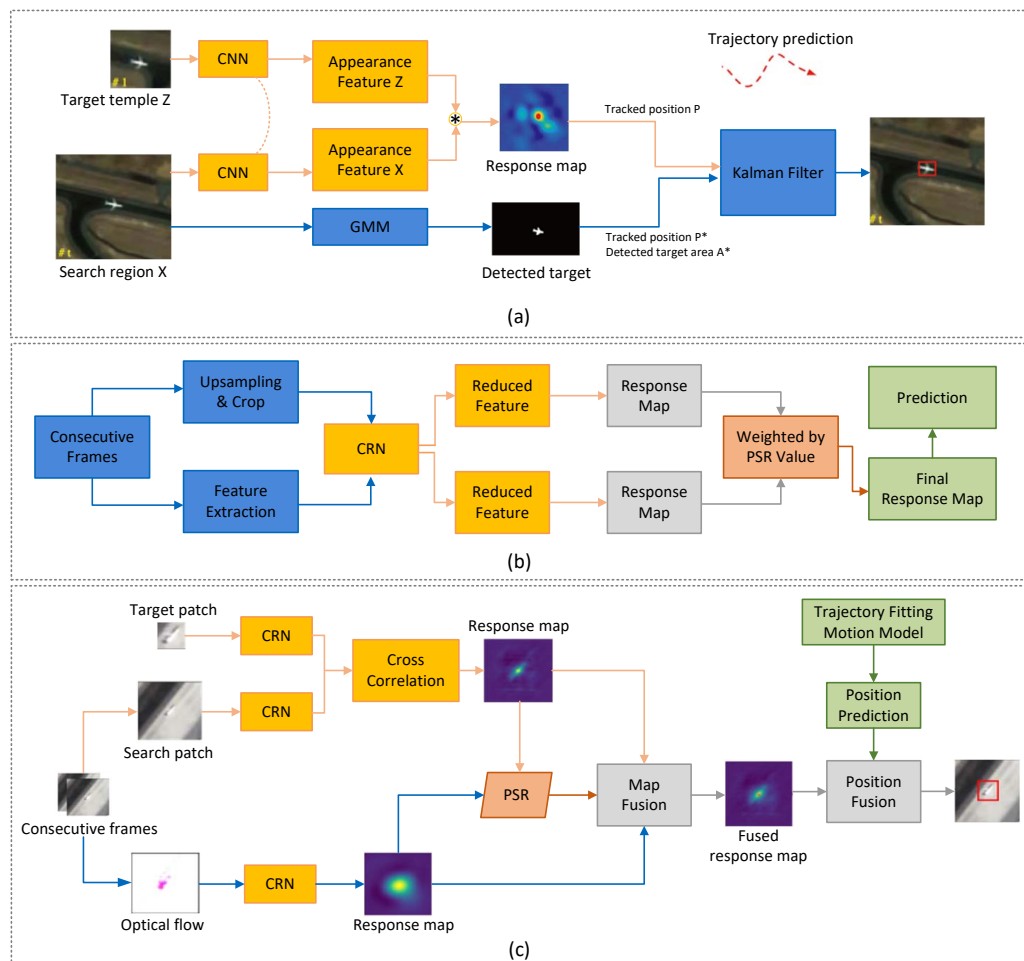

**Figure 3.** Comparison diagram of algorithm structure for DL-based traffic object tracking methods. (**a**) example of SN-based tracking method reproduced from Ref. [76]; (**b**) the overall structure of the CRAM (regression network (RN)-based) tracking network [79]; (**c**) the pipeline for the SN-RN combined tracking method reproduced from Ref. [80].

Recently, RNs have been studied and shown promising performance in the field of satellite video-based object tracking. For example, Ref. [79] introduced a convolutional regression network with appearance and motion feature (CRAM) (see Figure 3b), which consisted of training and tracking two phases. In the training phase, the two RNs were trained with different appearance and motion features respectively. In the tracking phase, the model responses were weighted by their qualities measured from the peak-to-sidelobe ratio (PSR) and then integrated for the final target location prediction [81]. To evaluate the performance of the proposed network, the authors collected nine small sequences with a total number of 31 moving vehicles, which were cropped from the SkySat-1 and Jilin-1 satellite videos. The average overlap measure and expected average overlap indices were analyzed, with the results of 0.7 and 0.7286, thereby demonstrating the efficiency of the presented network in object tracking from high-resolution remote sensing videos. Later, Ref. [82] suggested a cross-frame keypoint-based detection network based on a two-branch Long short-term memory (LSTM). The spatial information and motion information of moving targets are extracted for better tracking of the missed or occluded vehicles. Experimental results on Jilin-1 and SkySat satellite videos illustrated the effectiveness of the proposed tracking algorithms.

Furthermore, a prediction network (PN) was studied by [83], which predicted the location probability of the target in each pixel in the next frame using the fully convolutional network (FCN) learned from previous results. The authors further introduced a

segmentation method to generate the feasible region with an assigned high probability for the target in each frame. Experiments were carried out with nine satellite videos taken from the JiLin-1, indicating the superiority of the proposed method, as the author reported.

By taking advantage of both the SN and RN, Ref. [80] proposed a two-stream deep neural network (SRN) (see Figure 3c) that combined a SN and a motion RN for satellite object tracking. In Ref. [80], a trajectory fitting motion model (FTM) based on history trajectories was employed to further alleviate model drift. Comprehensive experiments demonstrated that their method performed favorably compared with the state-of-the-art tracking methods. Additionally, by exploring the temporal and spatial context, the object appearance model, and the motion vector from occluded targets, Ref. [84] designed a Reinforcement learning (RL)-based approach to enhance the tracking performance under complete occlusion. In addition, Ref. [85] explored the potential of graph convolution (GC) for multi-object tracking and modeled the satellite video tracking as a graph information reasoning procedure from the multitask learning perspective. Compared with state-of-the-art multi-object trackers, the tracking accuracy of this model increased by 20%.

To sum up, Table 4 illustrates recent published articles that study the DL-based tracking methods. As listed in Table 4, the SN-based models are widely utilized for object tracking in the remote sensing area. The CNN combined with CF tracking is another popular trend, which integrates the efficiency of the CF method and robustness of the CNN. Meanwhile, due to the advantages in time-series image processing, RN-based approaches have shown their potential for advanced tasks, such as long-term tracking or tracking with occlusion. Figure 3 then shows the frameworks of DL-based traffic object tracking among SN-based, RN-based, and SN-RN combined methods.

**Table 4.** Summary of DL-based traffic objects tracking methods.

| Method | Ref. | Year | Description |
|---|---|---|---|
| SN-based tracking | [76] | 2019 | Predicting attention-inspired SN |
|  | [77] | 2021 | DSN + ID-CIM |
|  | [78] | 2021 | Lightweight parallel network with a high spatial resolution |
| CNN combined with CF | [75] | 2018 | Kernelized CF utilizing deep CNN features |
| RN-based tracking | [79] | 2020 | CRAM, RN-based training |
| RN-based tracking | [82] | 2021 | A two-branch LSTM |
| PN-based tracking | [83] | 2021 | PN to predict the location probability of the target |
| Combined SN and RN | [80] | 2022 | SRN followed by FTM |
| RL-based tracking | [84] | 2022 | RL to track objects under occlusion |
| GC-based tracking | [85] | 2022 | Tracking via GC-based multitask reasoning |

### 3.1.4. Optical Flow-Based Methods

The optical flow method utilizes the apparent motion of the brightness patterns in the image to detect moving objects. The algorithm output can provide vital information for the tiny movements of an object [86]. It is worth noting that the background relative to the interested target is generally constant in satellite videos. Therefore, the image target and background can be separated by optical flow efficiently. If target objects move too slow to be analyzed with optical flow, multi-frame differences can be employed to improve the tracking performance [87]. Table 5 summarizes typical methods of the optical flow-based methods. Global feature-based optical flow is an old-fashioned method of tracking objects from remote sensing images, whereas local feature-based optical flow methods are gaining popularity recently. The general architecture of the optical flow based methods is depicted in Figure 2d.

**Table 5.** Summary of optical flow based traffic objects' tracking methods.

| Method | Ref. | Year | Description |
|---|---|---|---|
| Global feature-based | [88] | 2013 | Three-frame differencing scheme |
| Local feature-based | [22] | 2019 | Multi-frame optical flow tracker |
| | [89] | 2021 | SLIC + optical flow |
| | [90] | 2022 | HoG + optical flow |

Earlier researchers utilized a three-frame differencing scheme to detect and track vehicles globally. [88]. In Ref. [88], the authors firstly proposed a box filter to reduce the seam artifacts caused by considerable radiometric changes in different focal planes of the original stitched image. The grid was chosen such that tiles were approximately $1000 \times 1000$ pixels. The tile processors then enabled the global parallelism necessary to achieve real-time performance. In addition, the tile patches were further set up to overlap by about 80 pixels at each border to ensure that vehicles near the edges are included.

More recently, local feature-based methods were developed, and Ref. [22] implemented a multi-frame optical flow tracker to track the vehicles in satellite videos. The author first proposed a Lucas–Kanade optical flow method to obtain the optical flow field. The Hue-Saturation-Value (HSV) color system was then utilized to convert the two-dimensional optical flow field into a three-bands color image. Finally, the integral image was adapted to obtain the most probable position of the target. Five satellite videos provided by UrtheCast Corp. and Chang Guang Satellite Technology Co., Ltd. were applied in experiments, showing that the proposed method can track slightly moving objects accurately. Additionally, an optical flow motion estimation combined with a superpixel algorithm was presented by [89]. The authors used simple linear iterative clustering (SLIC) to realize superpixels, which made the object a more regular and compact shape. The output of the superpixel algorithm was then fed to the optical flow method to obtain and label the moving object. In 2022, Ref. [90] fused the histogram of oriented gradient (HoG) features and optical flow features to enhance the representation information of the targets. The author also developed a disruptor-aware mechanism to weaken the influence of background noise. Experimental results show that the proposed algorithm achieves high tracking performance with target occlusion.

### 3.2. Offline Tracking Methods

Online tracking can only use existing frames for tracking model updates, whereas offline tracking methods can benefit from all keyframes providing the smoothness constraint [91]. Since the satellite videos are generally downloaded from the aerial platform in advance, the offline video tracking models are implemented to entire video frames. Compared to the online video tracking algorithms, offline tracking is typically formulated as a global optimization problem to obtain the global optimal tracks. Furthermore, hyperspectral videos are usually introduced to improve the performance of the offline tracking models. Table 6 summarizes the reviewed offline traffic object tracking methods, which are divided depending on how many steps to obtain the tracking result. One-step-based methods utilize the tracker only, while two-step algorithms consist of both detector and tracker.

**Table 6.** Summary of offline traffic object tracking methods.

| Method | Ref. | Year | Description |
|---|---|---|---|
| One step-based | [92] | 2014 | Two paralleled trackers for initialization and tracking |
| | [93] | 2021 | 3D variation regularization + PCA |
| Two step-based | [94] | 2018 | Global data association approach |
| | [95] | 2019 | DTS for traffic parameters estimating |
| | [96] | 2019 | DTS for vehicle tracking |

In 2014, Ref. [92] proposed a fused framework for tracking multiple cars from satellite videos, in which two trackers worked in parallel. One tracker provided target initialization and reacquisition through detections from background subtraction. The other offered a frame to frame tracking by a target state regressor. A sequence from a publicly available wide-area aerial imagery dataset WPAFB was applied to test the proposed framework. Tracking metric indicators, namely, track swaps, track breaks, and overall MOTA, were calculated with 0.20, 0.92, and 0.41, respectively, in terms of detection and tracking metrics. Later, Ref. [93] incorporated a three-dimensional (3D) total variation regularization into the robust PCA model, in order to extract the moving targets from the background. Evaluation results on real remote sensing videos have demonstrated the advantage of this approach.

An offline two-step global data association approach was later presented in Ref. [94] to track multiple targets using satellite videos. The authors extended the spatial grid flow model to cover the possible connectivities in a wider temporal neighboring, making sure the association matches temporal-unlinked detections. Then, a KF-based tracklet transition probability was customized to link tracklets within large temporal intervals. To demonstrate traffic tracking capabilities, the proposed method was evaluated on a dataset that was cropped from a satellite high definition video captured by SkySat-1 on 25 March 2014.

On the other hand, Ref. [95] contributed to the integration of the two-step offline tracking algorithm, developing a complete and effective offline detection-tracking system (DTS) using satellite videos to estimate traffic parameters. In their system, a video preprocessing step is firstly applied to obtain the background. The moving targets were then checked over time to construct the target trajectories. A threshold method based on target displacement and velocity was utilized to eliminate false positives. A satellite video captured over Las Vegas from the SkySat-1 satellite with 30 FPS was applied to the proposed method. The results still revealed the limitation of the said method which was the inability of noise removal conditions to filter out tall buildings' relief displacement. Meanwhile, Ref. [96] offered an efficient DTS to track vehicles in multi-temporal remote sensing images. In the detection phase, the authors applied background subtraction, reduced searching space, and combined road prior information to improve detection accuracy. In the tracking phase, a dynamic association method under state judgment rules was designed to associate all potential target candidates. Additionally, a group dividing method was proposed to further improve the tracking accuracy. The proposed model was evaluated on a remote sensing video dataset with a 10 FPS frame rate and $4096 \times 2160$ pixels resolution. Completeness, Correctness, and Quality indices were utilized for the performance assessment with the results of 0.99, 0.97, and 0.97, showing the effectiveness of the presented method in tracking small vehicles from satellite sequences.

### 3.3. Discussion on Traffic Tracking Methods

To develop a general comparison, we summarize and elaborate on the strength and limitations of the reviewed tracking models, as shown in Table 7. For example, by utilizing the circulant matrix in the frequency domain to simplify the matrix inverse operation, effective tracking performance is achieved by correlation-based models. However, the occlusion and distractors can influence the tracking accuracy of the CF-based models. By contrast, the DL-based methods trained by extensive datasets improve the performance of the models in highly complex scenes. In addition, the DL model, as a good feature extractor, is flexible and able to integrate with CF models and TBD models. Considering the state-of-the-art works in general visual tracking tasks, such as Accurate Tracking by Overlap Maximization (ATOM) [97], SiamRPN [98], and GradNet [99], the DL and CF models show great potential for future development in satellite video tracking areas. The optical flow-based models require less memory and processing time because of effective alignment and optical flow algorithms, whereas they are sensitive to background noise. The TBD models consist of two steps: detection and tracking. The detection and tracking modules can be replaced by different algorithms separately, in which the tracking performance heavily depends on the detection modules.

**Table 7.** Summary of various satellite video based tracking methods for traffic objects.

| Methods | Advantage | Disadvantage | Prospect |
|---|---|---|---|
| CFs | - Circulant matrix to compute<br>- Low computing process<br>- Effective | - Unrobust to occlusion and distractor | High |
| DL | - Robust<br>- Good scalability<br>- High accuracy | - Require large dataset for training | High |
| TBD | - Adaptive to multi-targets<br>- Flexible backbones<br>- High accuracy | - Strongly depend on detection modules | Moderate |
| Optical flow | - Low processing time<br>- Low Memory cost | - Highly sensitive to noise | Low |

## 4. Ship Tracking

In recent years, ship detection and tracking have attracted a lot of attention in remote sensing because of the great potential in military application and port activities analysis. Compared with the vehicle targets, the size of the ship targets varies in a wide range, and the background of the track is commonly water, which may limit the performance of tracking methods. The feature of the water background is very similar to adjacent frames, which leads to ineffective motion information from the background analysis. Tracking algorithms such as optical flow-based tracker and offline tracking methods are thus not proper for ship tracking. Therefore, several novel models have been proposed to track ships from satellite videos.

In this section, we categorized the ship tracking approaches into two classes: image-based tracking methods and multi-modality-based tracking approaches. The summary of reviewed ship tracking publications is given in Table 8. In addition, Figure 4 shows a comparison of algorithm structure between two categories.

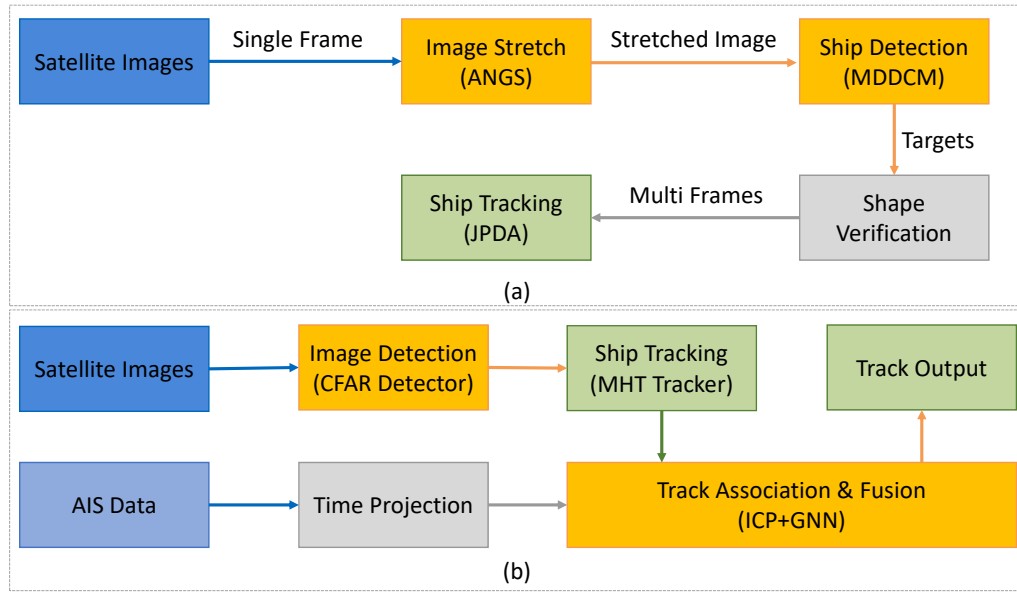

**Figure 4.** Comparison diagram of algorithm structure for ship tracking. (**a**) the framework of Ref. [100] (An Example of image-based tracking method); (**b**) the procedure of track-level fusion reproduced from Ref. [101] (An example of a multi-modality-based tracking method).

**Table 8.** Summary of the ship, typhoon, and fire tracking methods.

| Target | Method | Ref. | Year | Description |
|---|---|---|---|---|
| Ship | Image-based | [100] | 2019 | Automatic detection and tracking for moving ships |
| | | [102] | 2021 | Framework consists of ANGS, MDDCM, JPDA |
| | | [103] | 2022 | Mutual convolution SN with hierarchical double regression |
| | Multi-modality | [104] | 2010 | Ship detection and tracking using AIS and SAR data |
| | | [101] | 2018 | Track-level fusion for noncooperative ship tracking |
| | | [105] | 2018 | Integrate sequential imagery with AIS data |
| | | [106] | 2021 | Integrate satellite sequential imagery with ship location information |
| Typhoon | CNN-based | [107] | 2017 | A multi-layer model for multichannel image sequences |
| | | [108] | 2020 | A quasi-supervised mask region CNN |
| | GAN-based | [109] | 2019 | GAN to track and predict typhoon motion |
| | | [110] | 2021 | GAN with deep multi-scale frame prediction method |
| | | [111] | 2022 | GAN to predict both the track and intensity of typhoons |
| | RNN-based | [112] | 2017 | A convolutional sequence-to-sequence autoencoder |
| | | [113] | 2018 | MNNs for typhoon tracking |
| | | [114] | 2018 | A CLSTM based model |
| | | [115] | 2021 | A CLSTM layer with FCLs |
| | | [116] | 2022 | A CLSTM with 3D CNN based on multimodal data |
| | | [117] | 2022 | An echo state network-based tracking |
| Fire | Traditional | [118] | 2017 | Identify possible fire hotspots from two bands of AHI |
| | | [119] | 2018 | A threshold algorithm with visual interpretation |
| | | [120] | 2019 | A multi-temporal method of temperature estimation |
| | | [121] | 2020 | Temperature dynamics by data assimilation |
| | | [122] | 2022 | Wildfire tracking via visible and infrared image series |
| | DL-based | [123] | 2019 | 3D CNN to capture spatial and spectral patterns |
| | | [124] | 2019 | Inception-v3 model with transfer learning |
| | | [125] | 2021 | Near-real-time fire smoking prediction |
| | | [126] | 2022 | Combine the residual convolution and separable convolution to detect fire |
| | | [127] | 2022 | Multiple Kernel learning for various size fire detections |
| Ice motion | Traditional | [4] | 2017 | MCC tracker with hybrid example-based super-resolution model |
| | | [128] | 2017 | A faster cross-correlation based tracking with several updates |
| | | [129] | 2018 | A optical-flow based tracking with super-resolution enhancement |
| | | [130] | 2019 | A multi-step tracker for ice motion tracking |
| | | [131] | 2020 | Rotation-invariant ice floe tracking |
| | | [132] | 2021 | Integrating the cross-correlation with feature tracking |
| | | [133] | 2022 | Integrating locally consistent flow field filtering with cross-correlation |
| | DL-based | [134] | 2019 | An encoder-decoder network with LSTM to predict ice motion trajectory |
| | | [135] | 2021 | A CNN model to predict the arctic sea ice motions |
| | | [136] | 2021 | A multi-step machine learning approach to track icebergs |

### 4.1. Image-Based Tracking Methods

Ref. [100] developed an automatic detection and tracking model for moving ships in different sizes from satellite videos, as illustrated in Figure 4a. The dynamic multiscale saliency map was generated using motion compensation and multiscale differential saliency maps. Remote sensing images from the GO3S satellite were used to study the performance of the proposed method, indicating the effectiveness on ship tracking, especially on small ships. Furthermore, Ref. [102] proposed a new framework, including ANGS, MDDCM, and JPDA methods, to detect moving ships from GF-4 satellite images [137]. In Ref. [102], the ANGS enhanced the image and highlighted small and dim ship targets. The MDDCM detected the position of the candidate ship target, and the JPDA was applied for multi-frame data association and tracking. The authors analyzed that general influencing factors on ship detection in optical remote sensing images include bright clouds and islands. In addition, high-resolution images are encouraged for better detection scores. By designing

the mutual convolution Siamese network, Ref. [103] calculated the similarity between the object template and the search area to enhance the significance of the ship in the feature map. The authors also proposed a hierarchical double regression module to reduce the influence of the non-rigid motion of the water surface in the tracking phase.

### 4.2. Multi-Modality Based Tracking Methods

The AIS is an automatic tracking system that utilizes transceivers on ships and is applied by vessel traffic services. AIS information supplements marine radar, which continues to be the primary method of collision avoidance for water transport. AIS has been proven to be instrumental in accident investigation and search-and-rescue operations.

Earlier in 2010, Ref. [104] studied a fused ship detection and tracking system using the AIS data and satellite-borne SAR data. A 3D extension of a standard ordered-statistics constant false alarm rate (OSCFAR) algorithm was implemented on the radar data to realize target detection. For ship tracking, an alpha-beta filter combined with a nearest neighborhood assignment strategy was proposed and performed in polar coordinates to reduce false alarm errors. A time series of 512 samples and two onboard SAR sensors were used to verify their method, showing competitive results with previous works.

Recently, there has been renewed interest in fusing optical images with AIS data. Ref. [101] provided a track-level fusion architecture for GF-4 and AIS data to ship tracking tasks, as shown in Figure 4b. The constant false alarm rate (CFAR) detector first detected ships in GF-4 images, and then the multiple hypotheses tracking (MHT) Tracker with projected AIS data was aimed to achieve ship tracking. Then, the authors design a new track-to-track association algorithm based on iterative closest point (ICP) and global nearest neighbor (GNN) with multiple features to improve the validity of association. The core data fusion architecture was the track-to-track association based on a combined algorithm with multiple features to correct positioning errors. As reported by the authors, their effective data fusion method showed that the AIS aided satellite image offered a great perspective for tracking non-cooperative targets. Similar to Ref. [101], Ref. [105] investigated the AIS aided ship-tracking method with GF-4 satellite sequential imagery. The algorithm consisted of three steps: ship detection, position correction, and ship tracking, which were realized by the peak signal-to-noise ratio (PSNR)-based local visual saliency map, the rational polynomial coefficient (RPC) model with AIS data, and amplitude assisted MHT framework, respectively. The proposed method achieved the accuracy evaluation, precision, recall, and F1-score indices with 98.5%, 87.4%, and 92.6% on GF-4 satellite sequences, indicating the accurate estimation of moving ships. In 2021, Ref. [106] combined GOES-17 satellite imagery with ship location information to track the trajectories of ship-emitted aerosols based on its physical processes and optical flow model.

### 5. Typhoon Tracking

The rapid development of remote sensing technologies provides a new methodology for weather observation and forecasting tasks using high-resolution visual data [138]. Recently, a growing body of literature investigating the deep neural network-based cyclone track prediction from satellite imagery sequences has been published.

In this section, papers in the area of typhoon tracking methods are reviewed and divided into three classes, including the CNN-based models, GAN-based models, and RNN-based models, listed in Table 8. In addition, Figure 5 visualizes the three structures of CNN, GAN, and RNN-based typhoon tracking models.

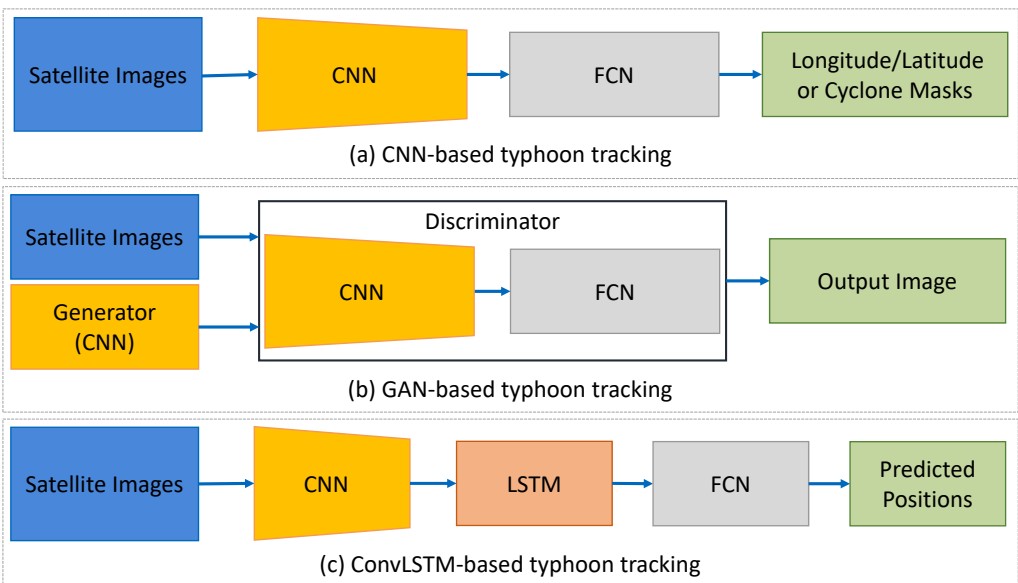

**Figure 5.** Comparison diagram of algorithm structure for (**a**) CNN-based, (**b**) GAN-based, and (**c**) RNN-based (specifically CLSTM) typhoon tracking.

## 5.1. CNN-Based Tracking Methods

To understand complex atmospheric dynamics based on multichannel 3D satellite image sequences, Ref. [107] introduced a multi-layer neural network. Multiple convolutional layers were first formed for typhoon feature extraction, followed by multiple fully connected dense layers with linear activation for linear metrics regression. In the regression step, the pixel related to the weather event was chosen as the target value. The proposed model was studied by a 2674-image satellite dataset acquired by the COMS-1 meteorological imagery [139], achieving a Root Mean Squared Error (RMSE) of ~0.02 to predict the center of a single typhoon that represented ~74.53 km in great circle distance. As the authors presented, a CNN could predict the coordinates of single typhoons efficiently, while the multiple typhoon case and unsupervised sequences of images needed to be further investigated. By further exploring the potential of the CNN models in cyclone detection, Ref. [108] designed a quasi-supervised mask region CNN. The seasonal march and spatial distribution of cyclone frequencies were derived from the proposed model. Compared with traditional methods, the presented method increased the number of identified cyclones by 8.29%, showing its good performance in identifying the horizontal structures of tropical cyclones.

## 5.2. GAN-Based Tracking Methods

Models such as those above can be categorized as discriminative models as they use conditional probability to predict the unseen data, while other methods employ generative models that make predictions by modeling joint distribution and are capable of generating new data. For example, Ref. [109] introduced a GAN to track and predict the typhoon centers and future cloud appearance simultaneously. A typical GAN structure was trained in an adversarial way to generate a 6-hour-advance track of a typhoon. The predicted typhoon track favorably identified the future typhoon location and the deformed cloud structures. The achieved averaged difference error between the predicted and ground truth typhoon centers was 95.6 km by calculating ten typhoon datasets. The tracking prediction could be significantly improved when employing both velocity fields and satellite images to deal with sudden changes in the track. Later, Ref. [110] integrated the GAN model with a deep multi-scale frame prediction algorithm, aiming to predict the atmospheric motion vectors of typhoons. The experiment results illustrated that the generated atmospheric motion vectors depicted the structure of typhoon atmospheric circulations with a certain

level of accuracy. Similarly, Ref. [111] designed a GAN based approach to predict both the track and intensity of typhoons for short lead times within fractions of a second. The experimental results indicated that learning velocity, temperature, pressure, and humidity along with satellite images have positive effects on trajectory prediction accuracy.

*5.3. RNN-Based Tracking Methods*

Another idea dealing with tracking tasks focuses on RNN models, which have shown promising performance in processing the time series data in various areas. Ref. [112] developed a convolutional sequence-to-sequence autoencoder in 2017 to predict the undiscovered weather situations from satellite image series. In 2018, Ref. [113] presented MNNs to predict cyclone tracks for satellite imagery sequences from the South Indian Ocean area. The MNNs were trained based on matrix convolutional units and utilized to propagate the information from the input matrix to the output layer. A dataset consisting of 286 cyclones was used to verify the effectiveness of the MNNs in typhoon tracking. In the same year, Ref. [114] designed a convolutional LSTM model to track and predict the tropical cyclone path. In their experiments, the proposed approach was successful in learning the spatiotemporal dynamics of the atmosphere.

In 2021, Ref. [115] compared various CNN and RNN recognition algorithms and proposed that the best performing network implemented a convolutional LSTM layer with FCLs. Cloud features rotating around a typhoon center were extracted by their model from the satellite infrared videos. Moreover, models trained with long-wave infrared channels outperformed a water vapor channel-based network. The average position across the two infrared networks has a 19.3 km median error across all intensities, which equated to a 42% lower error over a baseline technique. Later, by applying the multimodal data based on typhoon track data and satellite images, Ref. [116] integrated the LSTM and 3D CNN model to predict typhoon trajectory. In spite of widespread RNN structures, Ref. [117] studied an echo state network to track the typhoon based on the meteorological dataset, yet its potential for the image-based data still needs to be explored.

## 6. Fire Tracking

Fire tracking has become an attractive application of satellite remote sensing thanks to the characteristics of recent remote sensing images, such as high frequency, large range, and multi-spectrum. Additionally, the high-resolution images provide more information and high-time resolution data in forest fire monitoring, showing great potential in environment monitoring. In recent years, many researchers have concentrated on the activate fire detection based on single images, while a few pieces of literature tracked the fire and smoke based on multi-temporal detection or continuous detection. A vital component of fire tracking from remote sensors is the accurate estimation of the background temperature of an area in a fire's absence, which helps identify and report fire activity.

Therefore, this section provides a review of fire tracking methods and categorizes them into two classes, including the traditional methods and DL-based methods. A brief summary of the reviewed fire tracking methods can be seen in Table 8 and a comparison of two types of fire tracking methods can be seen in Figure 6.

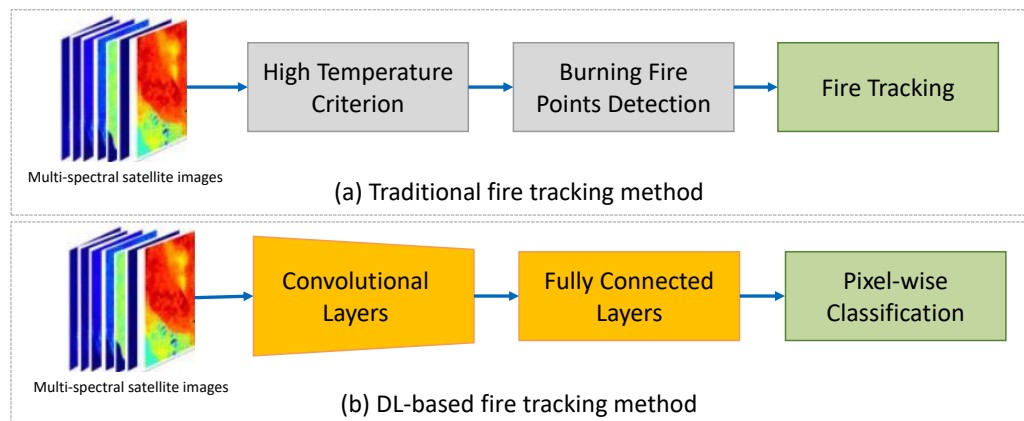

**Figure 6.** Comparison diagram of fire tracking algorithm structure for the (**a**) traditional method and the (**b**) DL-based method.

### 6.1. Traditional Tracking Methods

Regarding satellite imagery from satellite videos, important work for fire and smoke detection has been performed by applying the advanced AHI sensor of the Japanese geostationary weather satellite Himawari-8. The AHI offers extremely high-temporal-resolution (10 min) multispectral imagery, which is suitable for real-time wildfire monitoring on a large spatial and temporal scale.

Based on the AHI system, Ref. [118] investigated the feasibility of extracting real-time information about the spatial extents of wildfires. The algorithm first identified possible hotspots using the 3.9 μm and 11.2 μm bands of Himawari-8, and then eliminated false alarms by applying certain thresholds. A similar work was proposed in Ref. [119], which integrated a threshold algorithm and a visual interpretation method to monitor the entire process of grassland fires that occurred in the China-Mongolia border regions. To further explore the information from AHI image series, Ref. [120] extended their previous work and proposed a multi-temporal method of background temperature estimation. The proposed method involved a two-step process for geostationary data: a preprocessing step to aggregate the images from the AHI and a fitting step to apply a single value decomposition process for each individual pixel. Each decomposition feature map can then be compared to the raw brightness temperature data to identify thermal anomalies and track the active fire. Results showed the proposed method detected positive thermal anomalies in up to 99% of fire cases. Recently, Ref. [122] proposed a new object-based system for tracking the progression of individual fires via visible and infrared satellite image series. The designed system can update the attributes of each fire event in California during 2012–2020, delineate the fire perimeter, and identify the active fire front shortly after satellite data acquisition.

The previous methods can overestimate the background temperature of a fire pixel and, therefore, leads to the omission of a fire event. To address this problem, Ref. [121] designed an algorithm that assimilated brightness temperatures from infrared images and the offset of the sunrise to the thermal sunrise time of a non-fire condition. The introduction of assimilation strategies improved the data analysis quality and computational cost, resulting in better fire detection and tracking results.

### 6.2. DL-Based Tracking Methods

Instead of exploring the fire features via manually designed operators, Ref. [123] investigated DL-based remote wildfire detection and tracking framework from satellite image series. They firstly processed the streaming images to purify and examined raw image data to obtain ROI. Secondly, a 3D CNN was applied to capture spatial and spectral patterns for more accurate and robust detection. Finally, a streaming data visualization model was completed for potential wildfire incidents. The empirical evaluations highlighted that the

proposed CNN models outperformed the baselines with a 94% F1 score. To improve the fire detection accuracy, authors from [124] developed an effective approach of a CNN based Inception-v3 with transfer learning to train the satellite images and classify the datasets into the fire and non-fire images. The confusion matrix is introduced to specify the efficiency of the proposed model, and the fire occurred region is extracted based on a local binary pattern. More recently, Ref. [125] explored the potential of DL-based fire tracking by presenting a deep FCN to predict fire smoke, where satellite imagery in near-real-time by six bands images from the AHI sensor was used.

More DL-based methods contribute to fire detection instead of tracking. For example, Ref. [127] revised the general CNN models to enhance the fire detection performance in 2022. The proposed network consists of several convolution kernels with multiple sizes and dilated convolution layers with various dilation rates. Experimental results based on Landsat-8 satellite images revealed that the designed models could detect fires of varying sizes and shapes over challenging test samples, including the single fire pixels from the large fire zones. Similarly, Ref. [126] fused the optical and thermal modalities from the Landsat-8 images for a more effective fire representation. The proposed CNN model combined the residual convolution and separable convolution blocks to enable deeper features of the tracking target. A review of remote sensing-based fire detection is given in [140] in 2020, and more recent published works can be found in [141–143]. As detection is different from tracking and is out of our scope, we focus here on tracking only and do not provide the details on fire detection. Further studies could also be conducted to extend the DL-based fire detection to DL-based fire tracking.

## 7. Sea Ice Motion Tracking

Sea ice tracking is essential for many regional and local level applications, including modeling sea ice distribution, ocean atmosphere, climate dynamics, as well as safe navigation and sea operations. Most operational sea ice monitoring techniques rely on satellite-borne optical and SAR sensors, augmented by scatterometer and passive microwave imagery. In this review, previous ice tracking works are studied and classified into two categories: traditional tracking methods and DL-based tracking. Specifically, traditional ice tracking methods can be broadened to include cross correlation-based, optical flow-based, etc.

### 7.1. Traditional Ice Tracking Methods

In 2017, Ref. [4] utilized the maximum cross correlation (MCC) algorithm to estimate sea ice drift vectors and track the sea ice movements, in which a hybrid example-based super-resolution model was developed to enhance the image quality for better tracking performance. Meanwhile, Ref. [128] proposed several marked updates to speed up the cross-correlation-based algorithm. These updates include swapping the image order and matching direction, introducing a priori ice velocity information, and applying a post-processing algorithm. Experiment results revealed the improvement of the overall tracking performance based on cross-correlation. Later, Ref. [132] integrated the cross-correlation with feature tracking and proposed a fine-resolution hybrid sea ice tracking algorithm. The proposed method can be applied for regional fast ice mapping and large stamukhas detection to aid coastal research. Similarly, Ref. [133] designed a locally consistent flow field filtering algorithm with a correlation coefficient threshold and achieved better performance in sea ice motion estimation using GF-3 imagery.

Except for the cross correlation-based tracking, Ref. [129] introduced the optical flow algorithm to extract a dense motion vector field of the ice motion, achieving sub-pixel accuracy. An external example learning-based super-resolution method was applied to generate higher resolution tracking samples. This approach was successfully evaluated on the passive microwave, optical, and SAR, proving appropriate for multi-sensor applications and different spatial resolutions. Later, Ref. [130] proposed a multi-step tracker for ice motion tracking. By comparing ice floes within consecutive images, the algorithm extracted

the potential matches with thresholds and selected the best candidates based on the assessment of a similarity metric. The approach was utilized to track ice floes with length scales ranging from 8 km to 65 km from the East Greenland Current (ECG) for 6.5 weeks in spring 2017. Compared with manual annotations, the absolute position and tracking errors associated with the method were 255 m and 0.65 cm, respectively. Furthermore, authors from [131] designed a multi-step tracker for rotation-invariant ice floe tracking. Their approach consisted of ice floe extraction, ice floe description, and ice floe matching. The tracker enabled the identification of individual ice floes and the determination of their relative rotation from multiple Sentinel-2 images. Later, Ref. [144] combined an on-ice seismic network with TerraSAR-X satellite imagery to track the ice cracking from 2012 to 2014 in Pine Island Glacier. The author applied a flexural gravity wave model and deconvolved the wave propagation effects, implying that water flow may limit the rate of crevasse opening.

### 7.2. DL-Based Tracking

Compared with the various ice motion trackers based on traditional methods, DL-based approaches have been proposed in recent years for ice motion trajectory prediction. In 2019, Ref. [134] introduced an encoder-decoder network with LSTM units to predict sea ice motion in several days. The optical flow of ice motion, calculated from satellite passive microwave and scatterometer daily images, was fed to their network. According to the experiments, this method could forecast sea ice motion for up to 10 days in the future. Similarly, Ref. [135] established a CNN model and introduced previous day ice velocity, concentration, and present-day surface wind to track and predict the arctic sea ice motions. Results reveal that the designed CNN model computes the sea ice response with a correlation of 0.82 on average with respect to reality, which surpasses a set of local point-wise predictions and a leading thermodynamic-dynamical model. The ice motion tracking performance of CNN suggests the potential for combining DL with physics-based models to simulate sea ice. Later, Ref. [136] suggested a multi-step machine learning approach to track icebergs via SAR imagery. The proposed method consists of three stages, which are the graph-based superpixel segmentation model, the ensemble learning process with the heterogeneous model, and the incremental learning approach. The authors collect SAR satellite image series from the Weddell Sea region to verify the approaches. The experiment results show that the majority of the tracked icebergs drifted between 1.3 km and 2679.2 km westward around the Antarctic continent at an average drift speed of $3.6 \pm 7.4$ km/day.

Above all, the cross-correlation and optical flow algorithms play crucial roles in ice motion tracking. Integrating feature tracking with cross-correlation has been well studied and showed promising performance in ice motion tracking from remote sensing images. Furthermore, the success of the DL model in existing works suggests the feasibility and potential of combining machine learning with physics-based models to track and predict ice motion. However, considerably more work needs to be done to achieve competitive stability and accuracy in ice motion tracking compared with traditional methods.

## 8. Benchmark Dataset

A benchmark dataset is vital for tracking algorithm development and evaluation. Datasets from previous studies suggest that characteristics of different datasets can lead to different tracking strategies. We, therefore, discuss and summarize the available dataset for various tracking objects, and further develop a new dataset based on WPAFB for vehicle tracking.

### 8.1. Available Dataset

Many tracking algorithms have been employed for object tracking from satellite videos. However, higher tracking performance is constantly demanded. Compared with tracking algorithms in the traditional computer vision area, one of the major constraints of tracking performance in the remote sensing area is the limited dataset. Previous studies show that

several datasets for satellite tracking have been collected and introduced to provide fair and standardized evaluations of object tracking algorithms. We collect the dataset based on the standard of multiple reuses in different published works. In terms of tracking objects of the benchmark datasets, we divide the tracking benchmark datasets into two classes: artificial target datasets and natural target datasets. The artificial targets include vehicles, ships, trains, and planes, while the natural targets include typhoons, fire, and ice. According to the review results, there are four popular datasets for artificial satellite target tracking, and two datasets are collected for typhoon and fire tracking, respectively. Due to the limited existing literature, the public ice tracking dataset has not been found. The commonly-used satellite video datasets are detailed as follows.

1. **SatSOT dataset** [145]. The dataset focuses on satellite video single object tracking and comes from three commercial satellite sources: Jilin-1, Skybox, and Carbonite-2 satellites. Each raw video has a frame rate of 10 FPS or 25 FPS with about a 30 s duration. The 105 sequences of the dataset consist of 26 trains, 65 cars, nine planes, and five ships with an overall of 27,664 frames. Among the 105 sequences, 12 sequences with full occlusion are formed into a subset of long-term tracking. Compared with ships and planes, more car and train sequences are introduced. The average video length of SatSOT is 263 frames.

2. **VISO dataset** [146] This dataset is a large-scale dataset for moving object detection and tracking in satellite videos, which consists of 47 satellite videos captured by Jilin-1 satellite platforms. Each image has a resolution of 12,000 $\times$ 5000 pixel and contains a great number of objects with different scales. Four common types of vehicles, including planes, cars, ships, and trains, are manually-labeled. A total of 853,911 instances are labeled by axis-aligned bounding boxes.

3. **CVH dataset**. The Canada Vancouver harbor (CVH) dataset is a full color, ultra high definition (UHD) MPEG-4 file that has a spatial resolution of one meter, provided for the 2016 IEEE GRSS Data Fusion Contest by Deimos Imaging and Urthecast, acquired from International Space Station (ISS) high-resolution camera Irish on 2 July 2015 [147]. The dataset lasts 34 s and has 418 frames with the frame rate of being 27.97 FPS. The frame size is 3840 $\times$ 2160 pixel$^2$, covering an urban and harbor area in Vancouver, Canada, with an area of ~23.8 $\times$ 2.1 km$^2$.

4. **WPAFB dataset**. The wide-area aerial imagery dataset is taken by a camera system with six optical sensors and has already been stitched to cover a wide area of ~35 $\times$ 35 km$^2$. It is collected over the Dayton and Ohio area in October of 2009. This dataset contains 1025 frames with a 1.42 FPS frame rate. The input image size is averaged at 13,056 $\times$ 10,496 pixel$^2$ but changes from frame to frame, due to the orthorectify and stitch operations. More than 400 tracks of the vehicles in the dataset are labeled.

5. **JTWC dataset** [107]. The cyclone trajectory dataset is obtained from the Joint Typhoon Warning Center (JTWC) [148], which features the cyclones that occurred in the South Indian Ocean from 1985 to 2013. The dataset highlights 286 cyclones in total. The majority of the labeled cyclone duration lies between 20–40 time points, where each time point represents 6 h. The number of data points in each cyclone ranges from 6 to 129.

6. **Himawari-8 dataset** [149]. The Himawari-8 satellite is a Japanese weather satellite, operated by the Japan Meteorological Agency, and entered operational service on 7 July 2015. The satellite can provide observations every 10–30 min (with a higher spatial resolution 2 km pixel size that can be reduced to 500 m), making it ideal for near-real-time fire surveillance. Each image size of the Himawari-8 is 11,000 $\times$ 11,000 pixels$^2$, while the video length for each fire tracking is uncertain because of the large amount of the history images.

7. **MLTB**. To further develop moving target tracking, we design a multi-level tracking benchmark (MLTB) dataset based on the WPAFB dataset in terms of vehicle tracking. The details of data collection and sample processing will be discussed in Section 8.2.

The comparison between different datasets is shown in Table 9. Due to the different sizes and moving velocities of the tracking target, the resolution and FPS of different datasets are various. Generally, the dataset resolution and FPS utilized for vehicle tracking are relatively high, while for typhoon and fire are relatively low. Taking account of the FPS value, the WPAFB dataset has the highest one. A lower FPS refers to a larger time interval between adjacent frames, which indicates more difficulties in tracking the movement of the target. In addition, the frame rate of the typhoon-based tracking dataset is only 1/6 frames per hour because of the small velocity in low-resolution images.

**Table 9.** Satellite video dataset for object tracking.

| Dataset Name | Frame Size/Pixel | Frame Rate/FPS | Video Length | Labeled Target |
|:---:|:---:|:---:|:---:|:---:|
| SatSOT | $12{,}000 \times 5000$ | 10/25 | 263 frames | cars/ships/planes/trains |
| VISO | $12{,}000 \times 5000$ | 10 | ~30 s | cars/ships/planes/trains |
| CVH | $3840 \times 2160$ | 29.97 | ~30 s | vehicles/trains/ships |
| WPAFB | $13{,}056 \times 10{,}496$ | 1.42 | 1455 s | cars |
| JTWC | $512 \times 512$ | $4.6 \times 10^{-5}$ | ~774 h | cyclone trajectory |
| Himawari-8 | $11{,}000 \times 11{,}000$ | ~$8.3 \times 10^{-4}$ | / | fire |
| MLTB | $13{,}056 \times 10{,}496$ | 1.42 | 1455 s | cars |

The shortcomings of the existing satellite dataset for object tracking can be concluded by: (1) Compared with the general object tracking datasets in the computer vision area, the satellite datasets are relatively insufficient for performance evaluation. (2) Most of the vehicle tracking dataset is relatively short. Therefore, complex tracking situations are inadequate, such as illumination change, occlusions, and target motion change. (3) The WPAFB dataset is public and large tracking dataset for long-term tracking. Due to its low frame rate and occlusion scenes, tracking models can easily miss the target when it is occluded by trees or shadows. Even when the target appears again, it still fails to evaluate the model performance anymore. Therefore, this dataset is hard to apply in one tracking model by providing comparable prominent results.

*8.2. Dataset Processing*

For the future development of moving target tracking in the remote sensing area, we propose a MLTB dataset based on the WPAFB dataset in terms of vehicle tracking, as shown in Table 9. We carefully analyze each trajectory of all 401 tracks first and select the 184 tracklets with more than 100 frames to be our dataset. Then, we analyze difficult scenes in the dataset and categorize them into four classes, including occlusion, distractors, environment change, and target motion change. Specifically, the occlusion class includes the scenes where targets are occluded by trees, shadows, bridges, and buildings. Distractors class includes cross-roads or highway scenes where targets are close to other vehicles with similar appearance features. In environment change situations, the sudden change of illumination or light angle leads to a different apparent feature of tracking targets. Finally, the motion change class consists of scenes in which the targets suddenly stop, start, or change directions. The examples of the four categories are presented in Figure 7.

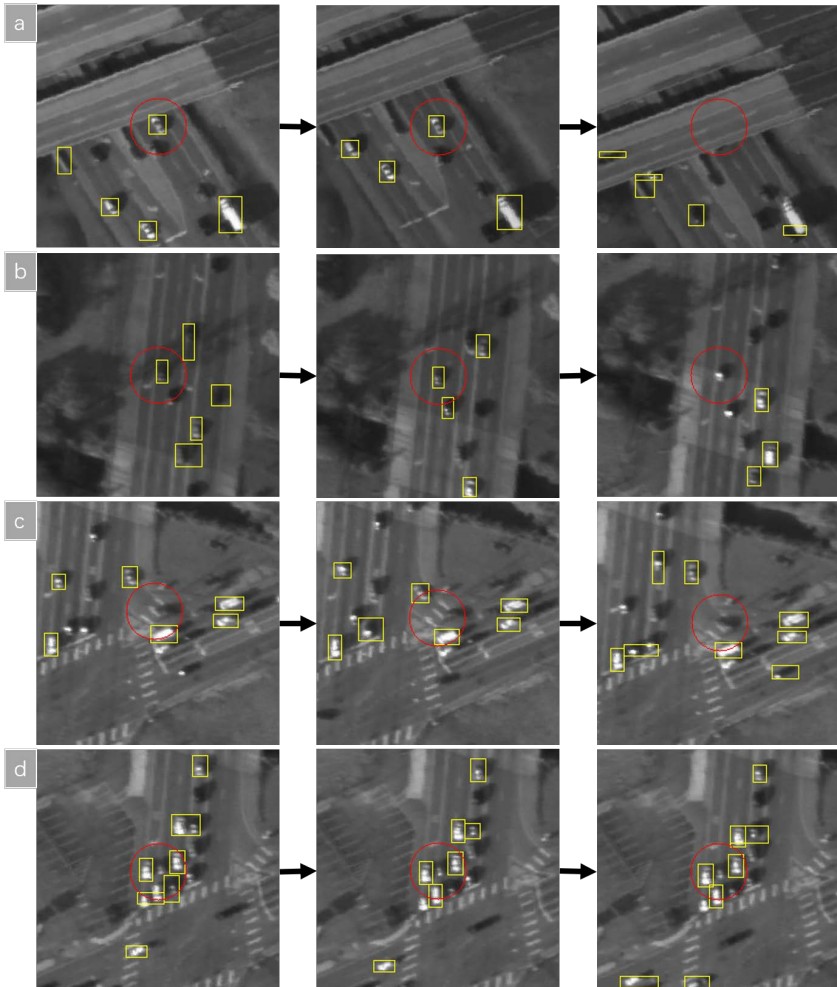

**Figure 7.** Four different categories of the proposed dataset. Center of red circle: targets. Yellow rectangle: Detection results. (**a**) occlusion; (**b**) environment change; (**c**) motion change; (**d**) distractors. (The original image is from the WPAFB dataset).

To accelerate the categorizing process, a DL-based vehicle detection model is introduced. To train the parameters of the detection model, we offer a remote sensing detection dataset UCAS-AOD dataset [150]. The UCAS-AOD Dataset is an open-source remote sensing image dataset, which contains two kinds of targets, automobiles and aircraft, and negative background samples. The detection benchmark UCAS-AOD is introduced as the training dataset. In addition, we cropped the traffic object samples from the WPAFB dataset and applied these samples as the test dataset, which contains 8871 samples. After testing several mainstream object detection algorithms on a sub-dataset based on the MMDetection framework [151], including YOLO [67], CenterNet [68], and CornerNet [69], we select CornerNet as the backbone model in our pipeline due to its good performance on small targets detection. It is worth mentioning that the purpose of vehicle detection is to distinguish the easy samples and hard samples, and the evaluation of detection performance is out of the scope of this work. The details of the proposed dataset annotation and implementation code are released in our Github repository github.com/caiya55/wpafb-dataset-relabeling (accessed on 10 July 2022).

The pipeline of the proposed dataset generation is explicated in Figure 8. As shown in Figure 8, the detection benchmark UCAS-AOD is firstly processed to train a CornerNet model. The preprocess module includes image cropping, histogram matching, and data augmentation. The pretrained CornerNet then detects each cropped patch from the WPAFB dataset. Next, the patches with ground-truth and detection results are manually evaluated

and categorized. This review designed the DS score to evaluate the quality of each positive sample in the proposed dataset, as shown in Equation (1):

$$DS = Occ + EC + 0.5 \times MC + 0.5 \times Dt \tag{1}$$

where Occ and EC indicate the target occlusion and environment changes, respectively. MC and Dt represent the motion changes and distractors, respectively. These factors for each target sample are manually annotated by three experts. The dimensionless DS is proposed to evaluate the label of the proposed dataset statistically, defined as the weighted sum of the four labels and delivered from Equation (1). Compared with the other two factors, the motion change and distractors have little effect on the tracking performance by analyzing the detection results and manual observation. Therefore, they are weighed by 0.5 in the DS metric. The distribution of the DS score for all tracklets is illustrated in Figure 9a. In Figure 9a, the tracklets are ranked by the DS score. Thus, the first 100 tracklets are selected and renamed the Easy group. From 100 to 150 tracklets, the corresponding tracklets are treated as the Medium group. The rest of the tracklets are grouped into the Hard group. The mean frames of the four categories for each group are demonstrated in Figure 9b. As shown in Figure 9b, the average frames of occlusion and environment change scene in each Easy tracklet are less than 8 and 3, respectively. By contrast, the average frames of the same situations in each Hard tracklet are more than 47 and 11, respectively. Hence, the general tracking methods can be evaluated and compared in the Easy group, which contains 100 tracklets and occasional occlusion scenarios. In addition, the tracklets in the Medium and Hard groups can evaluate tracking methods that are especially proposed for complex scenes, such as occlusion, plenty of distractors, and environment change.

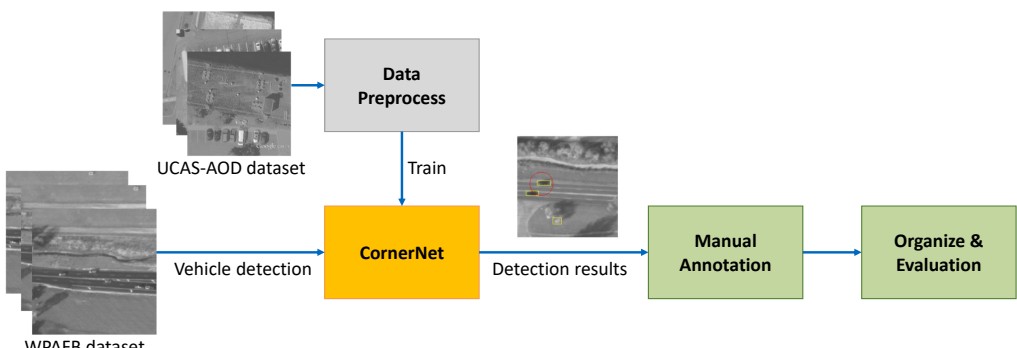

**Figure 8.** Pipeline of the proposed dataset generation.

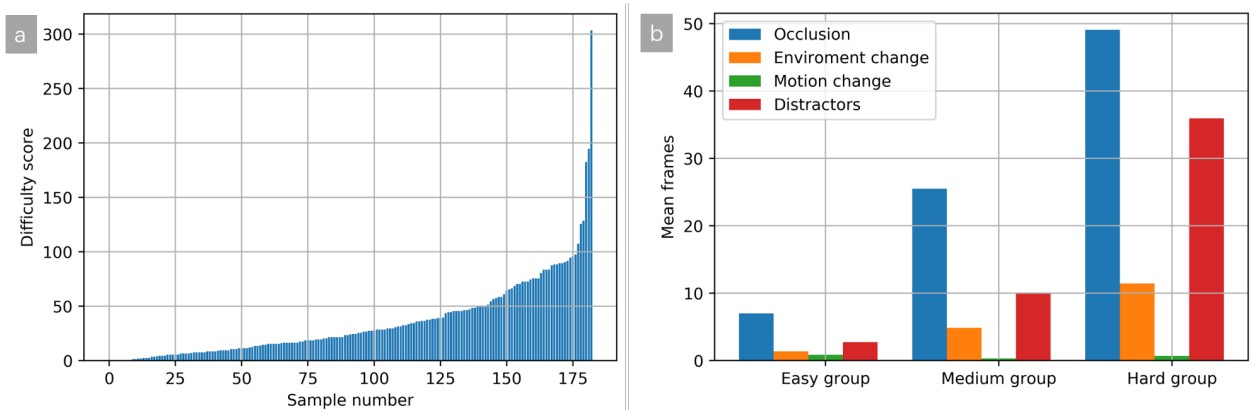

**Figure 9.** The distribution of categorized results in the proposed dataset. (**a**) the distribution of DS score; (**b**) the mean frames of the four categories for Hard, Medium, and Easy groups.

## 9. Conclusions and Future Directions

This paper reviews object tracking based on satellite videos for five major tracking objects. From a high-level perspective, the tracking objects and benchmark dataset are categorized into artificial targets (traffic objects and ships) and natural targets (typhoons, fire, and ice motion). The main differences between the artificial and natural targets are the motion velocity and size of the target, resulting in different spatial and temporal resolution datasets and various tracking algorithms. Specifically, high spatial resolution videos with high FPS are required to track vehicles, and furthermore, the multimodality data, such as AIS and SAR, are successfully integrated with the optical images to track cars and ships. Correspondingly, the available and suitable datasets for natural targets vary from the different sizes of objects. Since the large scale of the typhoon target, the multi-temporal low-resolution remote sensing dataset with low FPS is popular with typhoon tracking and trajectory predicting, while the AHI sensor and its dataset with extremely high-temporal-resolution multispectral imagery capability dominate the fire tracking area. As for the ice motion tracking, medium-resolution images with a large field of view are suitable.

In terms of tracking techniques, traffic object tracking has been widely studied due to its great societal, economic, and military value. From a high-level perspective, online and offline tracking methods are reviewed, and the online algorithms are further divided into CF-based, TBD, DL-based, and optical flow-based methods. For typhoon tracking, the DL-based framework has shown great promise, especially for predicting cloud appearance and typhoon centers using GAN. The tracking for fire benefits from the background temperature estimation-based traditional approach and provides a simple yet effective way to track the wildfire. Furthermore, the DL-based models provide better fire tracking with better robustness and accuracy, and more research should be conducted on extending DL-based fire detection to fire tracking. As for ice motion tracking, traditional methods, such as cross-correlation and optical flow algorithms, play crucial roles in this area. Moreover, the success of the DL model in existing works suggests the feasibility and potential of combining DL with physics-based models to track and predict ice motion. To sum up, traditional tracking methods have been studied widely and prove to be effective in tracking a variety of targets, while the DL-based approach is increasingly popular in tracking remote sensing objects and can extract complex features from backgrounds.

Remarkable developments in remote sensing imaging-based object tracking have been studied, yet research to date still has bottlenecks. One of the primary issues is occlusion, where targets may get lost in view during occlusion, and tracking models may not resume tracking when occlusion ends. Another issue is the changing target appearance caused by different atmospheric environments and illumination conditions. Several algorithms, such as motion estimation methods, tracklet association models, and DL-based trackers, have been investigated to sort out the above challenges, but more effort is needed. Furthermore, to achieve better accuracy, the tracking models integrated with other data sources, such as Global Positioning System (GPS) data, digital elevation model (DEM), and SAR data, would be a fruitful area for further work.

**Author Contributions:** All authors have contributed to this review paper. Z.Z. initiated the review and acquired funding, contributed to writing, identified selected research to include in the review, developed the proposed dataset, and coordinated input from other authors. J.S. contributed to writing, editing, and overall context organization. C.W. contributed to related publication searching and editing. Y.X. contributed to identifying selected research to include in the review. Z.Z. and J.S. have revised the manuscript. All authors have read and agreed to the published version of the manuscript.

**Funding:** The project was supported partially by China National Funds for Distinguished Young Scientists and Natural Science Basic Research Program of Shaanxi, under Grant No. D5110220135. The APC was funded by Fundamental Research Funds for the Central Universities, under Grant No. D5000210767.

**Institutional Review Board Statement:** Not applicable.

**Informed Consent Statement:** Not applicable.

**Data Availability Statement:** The proposed dataset annotation and implementation code are released in our Github repository and are accessible at github.com/caiya55/wpafb-dataset-relabeling.

**Acknowledgments:** We thank the anonymous reviewers and editors for their constructive comments and suggestions, which helped us to improve the manuscript. Our thanks also go to all those who shared their knowledge, publications, and studies selflessly.

**Conflicts of Interest:** The authors declare no conflict of interest.

## Abbreviations

The following abbreviations (ordered alphabetically) are used in this article:

| | |
|---|---|
| 2D | two-dimensional |
| 3D | three-dimensional |
| AIS | automatic identification system |
| AKF | adaptive Kalman filter |
| AHI | advanced Himawari imager |
| ANGS | adaptive nonlinear gray stretch |
| AUC | area under curve |
| B/T | breaks per track |
| CLE | center location error |
| CF | correlation filter |
| CFAR | constant false alarm rate |
| C-GICA | Cumulative Geometrical Independent Component Analysis |
| CNN | convolutional neural network |
| CLSTM | Convolutional LSTM |
| CRAM | convolutional regression network with appearance and motion feature |
| CSRT | Channel and Spatial Reliability Tracker |
| CVH | Canada Vancouver harbor |
| DCF | discriminative correlation filters |
| DEM | digital elevation model |
| DL | deep learning |
| DMSM | dynamic multiscale saliency map |
| DS | Difficulty Score |
| DSN | deep Siamese network |
| DTS | detection-tracking system |
| EAO | expected average overlap |
| ECOT | Efficient Convolution Operator Tracker |
| ECG | East Greenland Current |
| FCL | fully connected layer |
| FCN | fully convolutional network |
| FTM | fitting motion model |
| FPS | frame per second |
| GAN | Generative Adversarial Network |
| GC | graph convolution |
| GMM | Gaussian Mixture Model |
| GICA | Geometrical Independent Component Analysis |
| GPS | Global Positioning System |
| GNN | global nearest neighbor |
| GRU | gated recurrent unit |
| GSD | Ground Sampling Distance |
| HRSiam | High-resolution Siamese network |
| HCF | Hierarchical Convolutional Features |

| | |
|---|---|
| HLT | hyperspectral likelihood maps-aided tracking |
| HoG | histogram of oriented gradient |
| HSV | Hue–Saturation–Value |
| ID-CIM | interframe difference centroid inertia motion |
| ISS | International Space Station |
| ICP | iterative closest point |
| JPDA | joint probability data association |
| JTWC | Joint Typhoon Warning Center |
| KF | Kalman filter |
| KCF | kernel correlation tracker |
| LEO | Low Earth Orbiting |
| LSTM | Long short-term memory |
| MLTB | multi-level tracking benchmark |
| MCC | maximum cross correlation |
| MNN | matrix neural network |
| MDDCM | multiscale dual-neighbor difference contrast measure |
| MHT | multiple hypotheses tracking |
| ML | machine learning |
| ME | motion estimation |
| MOTA | Multiple Object Tracking Accuracy |
| MOTP | Multiple Object Tracking Precision |
| NMS | nonmaximum suppression |
| OSCFAR | ordered-statistics constant false alarm rate |
| PCA | principal component analysis |
| PSR | peak-to-sidelobe ratio |
| PSNR | peak signal-to-noise ratio |
| PN | prediction network |
| RGB | Red-Green-Blue |
| RL | Reinforcement learning |
| ROI | region of interest |
| RPC | rational polynomial coefficient |
| RN | regression network |
| RNN | recurrent neural network |
| RMSE | Root Mean Squared Error |
| SAR | synthetic aperture radar |
| SFMFT | slow feature and motion feature-guided multi-object tracking |
| SLIC | simple linear iterative clustering |
| SRN | two-stream deep neural network |
| SN | Siamese network |
| TADS | Target-awareness and Depthwise Separability |
| TCM | tracking confidence module |
| TBD | tracking-by-detection |
| UAV | unmanned aerial vehicle |
| UHD | ultra high definition |
| VCF | velocity correlation filter |
| VHR | very high resolution |
| WoS | Web of Science |
| WPAFB | Wright Patterson Air Force Base |

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
