# Peer review of "Object Tracking Based on Satellite Videos: A Literature Review"

_remotesensing, doi:10.3390/rs14153674_

Round 1
Reviewer 1 Report
This paper made a comprehensive review of the satellite video-based target tracking methods, which is a challenging and inspiring work in this area. Considering various tracking objects, the review paper logically introduce 5 typical scenarios along with their related popular algorithms, including Traffic objects tracking, Ship tracking, Typhoon tracking, Fire tracking and Sea ice motion tracking. Moreover, benchmark datasets are investigated and a multi-level satellite dataset is generated for further evaluation.
The contents of satellite object tracking methods in the review are comprehensive and complete with enough literature. However, this review still has some problems to be solved before final accepting.
1) In line 198, the figure is not referenced correctly. The reference seems incorrect.
2) In the field of ship tracking, why does water as background affects the performance of tracking method presented in line 510?
3) For the evaluation of the revised dataset, the selected backbone CornerNet is not introduced in the previous review, and the reason for choosing it is not fully explored. More algorithms are suggested to test on the proposed dataset.
4) The available datasets listed in section 8.1 are not classified by the categories of tracking targets discussed in section 2. In addition, in order to better present the proposed dataset UCAS-AOD, an introduction of it could be added into table 8.
5) All of the figures can be promoted into a high-resolution format, for example, Fig. 3.
Reviewer 2 Report
The key contributions claimed by the authors of this manuscript:
“This paper reviews the object tracking based on satellite videos for five major tracking objects, including the traffic targets, ship, typhoon, fires, and ice. A broad taxonomy of state-of-the-art tracking approaches is developed based on tracking targets, tracking models, and data modalities. The motivations and contributions of the reviewed methods are summarized and demonstrated by the tracking objects and tracking solutions. Existing benchmark datasets are compared and discussed. Furthermore, a multi-level visual tracking dataset is prepared and quantitatively analyzed.”
The English writing, presentation, and organization of this survey are good. The abstract is well-written but the conclusion must be revised. It is a good research survey with some research contributions, containing very updated reference literature. There are several concerns which authors must address:
1. Thank you for providing a list of acronyms. It’s really helpful to enhance better readability.
2. Figures are nicely presented with complete labeling and sufficient discussion.
3. Tables 1-5 do not provide a comprehensive description. The authors are suggested to provide sufficient details.
4. Typo or missing Figure number in line 198: Fig. ??.
5. Figure 1. The tree diagram is nicely presented.
6. It is highly recommended to provide a Table to evaluate your research contributions by comparing with existing reviews or surveys.
7. Line 110-111: It will be better if you can add proper references for discussed applications. Same for 144-145
8. Line 179-183: I have several concerns about it. 1st how many articles did you collect by using these keywords? 2nd why did you use only WoS and Google Scholar? How about other databases you might have missed some good articles in this way. 3rd what was the time span? Where are inclusion and exclusion criteria?
9. Until you do not provide complete details about the above comment, you cannot use the word Systematic for this survey. Better to check Systematic literature reviews and check the organization and presentation of those studies.
10. At the start of each section, please provide a short summary to introduce each section.
11. Table 7: You have missed some reported studies, especially for ML/DL. You can simply use Google Scholar for confirmation.
12. Insufficient discussion is provided in Subsections 6.2 and 7.2.
13. I suggest making the conclusion and future directions (section 9) more precise and short.
14. For reference section: Although it contains updated references mostly from the recent 5 years, but you have missed several reported studies on this topic.
Reviewer 3 Report
Authors contributions:
Satellite video-based visual tracking technologies are classified based on their monitoring goals, tracking network training, and network tracking.
The existing satellite video benchmark datasets are compared and analyzed.
A revised multi-level dataset with manual annotation is constructed, and quantitative and qualitative experimental evaluations for the aforementioned dataset are presented.
I have some reviewer notes:
Abstract. What is the accuracy from the qualitative evaluation of datasets?
Introduction part. If there are more applications of visual object tracking, you have to note them. And make clear description why you make survey only on seven of them.
Figure 7. If the images are from other literature sources, you have to cite them.
4.2. Multi-modality based tracking methods. Are there any other fusion methods for ship tracking? Also you can cite more papers about data fusion in tracking ice, fire etc.
8.1. Available dataset. If it is possible, it will be good to show geographical coordinates where the images in these datasets are.
Equation (1). If “DS” is dimensionless, you have to describe it in the text.
Conclusion part. How the proposed dataset improves the known solutions in this study area?
I have some suggestions:
Present your results with values. Make more comparative analyses with other papers. These suggestions will improve your contribution.
Reviewer 4 Report
This article presents a very interesting review of the object tracking based on satellite videos for five major tracking objects. These tracking objects are traffic targets, ships, typhoons, fires, and ice. Based on tracking targets, tracking models, and data modalities, a comprehensive taxonomy of state-of-the-art tracking approaches is developed. Comparison and discussion are done on the existing benchmark datasets. In addition to this, a multi-level visual tracking dataset is constructed and then subjected to quantitative analysis.
Tracking objects and benchmark datasets are divided into artificial targets (vehicles, ships, trains, and planes) and natural targets (typhoons, fire, and ice tracking). These two types of targets are distinct from one another in terms of the motion velocity and size of the target, which leads to a wide variety of spatial and temporal resolution datasets and tracking algorithms.
According to the research that has been done up to this point, traffic targets are the most common type of tracking object. This is likely due to the great societal, economic, and military value that they possess. In order to successfully track cars and ships, the data from multiple modalities, such as AIS and SAR, are successfully integrated with the optical images. The multi-temporal low-resolution remote sensing dataset with low frames per second (FPS) is becoming increasingly popular for use in typhoon tracking and trajectory predicting due to the large scale of the typhoon target. GAN has demonstrated significant potential in predicting the centers of hurricanes and the appearance of clouds in the future. The AHI sensor and its dataset, which have the capability to produce extremely high-temporal-resolution multispectral imagery, cover the area where the fire is being tracked. Traditional approaches that are based on an estimation of the background temperature offer a method that is straightforward and efficient for tracking the wildfire, whereas DL-based models have the advantage of being more accurate and robust when it comes to tracking the wildfire. It has been extensively researched, and the results have shown that integrating feature tracking with cross-correlation can produce promising results in ice motion tracking. In addition, the success of the DL model in previous works hints at the possibility and potential of combining machine learning with models based on physics in order to track and predict the motion of ice.
Because of the remarkable progress made in remote sensing imaging technologies, the various tracking methods now have distinct algorithms for the extraction of target features, tracking in low frames, and tracking in multispectral imagery, respectively. However, there are still "bottlenecks" that prevent further progress in satellite video-based tracking. Occlusion is one of the most significant problems, as it can cause targets to become hidden from view and prevent tracking models from resuming their work after a period of occlusion has ended. Several different algorithms, like motion estimation methods and tracklet association models, show promise as potential solutions to this problem. Alterations in the appearance of the target brought on by varying atmospheric environments and lighting conditions constitute a second problem. As one of the most prominent feature extractors, CNN has the ability to interpret target-related scenes, recognize local background structures, and effectively infer existing objects. In addition, the architecture that is based on Siamese can be altered to lessen the computational complexity of DL-based trackers and can incorporate these trackers with other tracking strategies. In spite of the fact that dozens of remote sensing satellites are put into orbit every year, inadequate tracking data has always been a fundamental obstacle for tracking performance. Tracking models that are integrated with other data sources, such as Global Navigation Satellite Systems (GNSS) data, digital elevation model (DEM) data, and SAR data, are thoroughly researched in order to improve the accuracy of the tracking results.
The review overall seems very well written, complete and worthy of publication. However, I suggest a re-reading to eliminate some (very few) remaining imperfections, as:
- Lines 7, 175. Four applications are indicated, while throughout the text and in Figure 1 five different applications are illustrated as tracking applications: traffic target tracking, ship tracking, typhoon tracking, fire tracking and ice motion tracking. On line 807, the indication is correct.
- Line 198: please, replace "??" with "Figure 1".
- Lines 515, 518, 542, 563, 623, 781, etc.: as stated in the template, all figures and tables should be cited in the main text as Figure 1, Table 1, etc.
- Line 512: remove the dot after the word Table.
Congratulations, it seems like a really good job!
Round 2
Reviewer 2 Report
Dear authors,
Thank you very much for your detailed response. I highly appreciate your response letter and efforts to improve the quality of this work. For me, it is acceptable for publication.